# PREFERENCE-BASED POLICY OPTIMIZATION FROM SPARSE-REWARD OFFLINE DATASET

**Wenjie Qiu**[1]    **Guofeng Cui**[1]    **Shicheng Liu**[2]    **Yuanlin Duan**[1]    **He Zhu**[1]

[1]Department of Computer Science, Rutgers University
[2]Department of Electrical Engineering, Pennsylvania State University
`{wq37,gc669,yuanlin.duan,hz375}@rutgers.edu`

## ABSTRACT

Offline reinforcement learning (RL) holds the promise of training effective policies from static datasets without the need for costly online interactions. However, offline RL faces key limitations, most notably the challenge of generalizing to unseen or infrequently encountered state-action pairs. When a value function is learned from limited data in sparse-reward environments, it can become overly optimistic about parts of the space that are poorly represented, leading to unreliable value estimates and degraded policy quality. To address these challenges, we introduce a novel approach based on contrastive preference learning that bypasses direct value function estimation. Our method trains policies by contrasting successful demonstrations with failure behaviors present in the dataset, as well as synthetic behaviors generated outside the support of the dataset distribution. This contrastive formulation mitigates overestimation bias and improves robustness in offline learning. Empirical results on challenging sparse-reward offline RL benchmarks show that our method substantially outperforms existing state-of-the-art baselines in both learning efficiency and final performance.

## 1 INTRODUCTION

Offline reinforcement learning (RL) (Levine et al., 2020; Prudencio et al., 2024) aims to learn high-quality decision policies purely from static datasets, without requiring additional environment interactions. This paradigm offers a compelling route for deploying RL in real-world domains where data collection is costly, risky, or constrained—such as robotics, healthcare, or recommendation systems. However, offline RL remains fundamentally challenging due to the distributional mismatch between the policy being learned and the limited data it learns from.

A core issue lies in the extrapolation error that arises when learned value functions are queried on state-action pairs not well represented in the dataset. This is particularly problematic in sparse-reward settings, where the dataset may lack sufficient reward-bearing trajectories or behavioral diversity. As a result, value-based methods can become overly optimistic in poorly covered regions of the state-action space, leading to unstable or suboptimal policies (Levine et al., 2020).

To address this, prior work has largely focused on three classes of solutions. First, pessimism-based approaches mitigate overestimation by explicitly penalizing uncertain or unsupported regions in the learned value function. Techniques such as conservative Q-learning (Kumar et al., 2020) or uncertainty-aware backups enforce value suppression on out-of-distribution actions. However, these methods often rely on assumptions about the behavior policy and require careful calibration of the degree of pessimism, which becomes increasingly difficult in high-dimensional or sparse settings (Liu et al., 2020; Xie et al., 2021). Second, regularization-based methods constrain policy updates to remain close to the behavior policy by adding policy divergence penalties (Wu et al., 2019; Kumar et al., 2019; Fujimoto & Gu, 2021). While effective in well-covered datasets, these methods can be brittle when tuning the regularization strength and may fail to explore beyond suboptimal behaviors (Lee et al., 2021; Brandfonbrener et al., 2021; Lyu et al., 2022). Third, importance sampling-based techniques, including DICE-style distribution correction (Cen et al., 2024; Lee et al., 2021), attempt to re-weight observed rewards based on estimated marginal state-action densities. Although theoretically sound and behavior-agnostic, these methods are sensitive to support mismatches and can suffer from

high variance or instability, especially when the data are limited or reward signals are sparse (Xie et al., 2021; Li et al., 2022; Cen et al., 2024).

In this paper, we propose a fundamentally different approach that avoids direct value function estimation altogether. We introduce **PREFORL** (PREFerence-based Optimization for Offline RL), a contrastive preference learning framework that optimizes policies by comparing successful (preferred) and unsuccessful (nonpreferred) behaviors from a static dataset. Previous work such as CPL (Hejna et al., 2024) and DPPO (An et al., 2023) has considered this setting, although not explicitly in the offline RL context. Nonetheless, simply contrasting successful and unsuccessful behaviors does not resolve overestimation in datasets with limited state–action coverage, since the policy can still become overly optimistic in poorly represented regions due to the absence of strong counterexamples. Crucially, in PREFORL, we extend the contrastive signal beyond failure behaviors present in the dataset to include synthetic behaviors generated outside the dataset's support. By contrasting both types of behaviors against successful demonstrations, our method trains policies to imitate not just what succeeds, but to actively avoid what likely fails or lies outside the dataset's support.

This formulation enables us to sidestep the estimation pitfalls of value-based methods while directly combating overestimation. Our empirical evaluation on challenging sparse-reward offline RL benchmarks shows that this contrastive approach leads to more stable learning and substantially outperforms existing state-of-the-art offline RL baselines.

## 2 PROBLEM FORMULATION AND MOTIVATIONS

We formulate the reinforcement learning problem in the context of a Markov Decision Process (MDP) $M = \langle \mathcal{S}, \mathcal{A}, T, r, \gamma, \rho_0 \rangle$, where $\mathcal{S}$ is the state space, $\mathcal{A}$ is the action space, $T : \mathcal{S} \times \mathcal{A} \times \mathcal{S} \rightarrow [0,1]$ is the transition probability function $T(s' \mid s, a)$, $r : \mathcal{S} \times \mathcal{A} \rightarrow \mathbb{R}$ is the reward function, $\gamma \in (0,1)$ is the discount factor, and $\rho_0 : \mathcal{S} \rightarrow [0,1]$ is the initial state distribution. A policy $\pi : \mathcal{S} \times \mathcal{A} \rightarrow [0,1]$ maps each state to a distribution over actions. Let $\tau = \{s_0, a_0, s_1, a_1, \ldots\}$ denote a trajectory sampled by interacting with the MDP under policy $\pi$, i.e., $s_0 \sim \rho_0$, $a_t \sim \pi(\cdot \mid s_t)$, $s_{t+1} \sim T(\cdot \mid s_t, a_t)$. Then, the discounted state-action distribution induced by $\pi$ is defined as $d^\pi(s, a) = (1 - \gamma) \sum_{t=0}^{\infty} \gamma^t \mathbb{E}_{\tau \sim \pi} [\mathbb{1}[s_t = s, a_t = a]]$. The goal is to learn a policy $\pi_\theta(a|s)$ that maximizes the expected discounted return: $\mathbb{E}_{s_0, a_0, s_1, \ldots \sim d^{\pi_\theta}} [\sum_0^\infty \gamma^t r(s_t, a_t)]$. In offline RL, the agent does not have access to the environment $M$, and instead must learn a policy solely from a static dataset $\mathcal{D}$ collected from some (possibly unknown) behavior policy $\pi_\beta$. We define $\mathcal{D} = \bigcup_{i=1}^{N} \tau_i$, where $\tau_i = \{(s_t^{(i)}, a_t^{(i)}, r_t^{(i)}, s_{t+1}^{(i)})\}_{t=1}^{T_i}$, with $N$ trajectories in total and $T_i$ denoting the length of the $i$-th trajectory. The empirical state-action distribution of the dataset is denoted $d^{\mathcal{D}}(s, a)$, which approximates $d^{\pi_\beta}(s, a)$. We consider the challenging setting of *sparse reward offline RL*, where informative reward signals are infrequent and the dataset $\mathcal{D}$ predominantly consists of transitions with zero or low rewards, making it difficult to identify and generalize from successful behaviors. Formally, we assume $\mathcal{D} = (\mathcal{D}^+, \mathcal{D}^-)$, where $\mathcal{D}^+$ contains successful trajectories and $\mathcal{D}^-$ contains unsuccessful trajectories. For a trajectory $\tau = \{(s_t, a_t, r_t, s_{t+1})\}_{t=1}^{T}$, let $R(\tau) = \sum_{t=1}^{T} r_t$ denote its cumulative return. We define $\mathcal{D}^+ = \{\tau \in \mathcal{D} \mid R(\tau) > \eta\}$, and $\mathcal{D}^- = \{\tau \in \mathcal{D} \mid R(\tau) \leq \eta\}$, where $\eta$ is a threshold. For example, in many sparse-reward environments $\eta = 0$, since trajectories that terminate in goal states receive a positive terminal reward, while those that do not yield zero cumulative return. We define $d^*(s)$ as the optimal state marginal, which can be viewed as a state distribution of successful trajectories $\mathcal{D}^+$ in the dataset $\mathcal{D}$.

**The Advantage Preference Model.** In Direct Preference Optimization (DPO) (Rafailov et al., 2023), a Bradley-Terry (BT) (Bradley & Terry, 1952) model is built on top of the hidden reward model $r_E$ given by expert users to capture the preferences of pairs of answers $(y_1, y_2) \sim \pi_\theta(y|x)$. While DPO and other reinforcement learning from human feedback (RLHF) algorithms (Christiano et al., 2017) have shown strong performance for large language models (LLMs)—which can be framed as contextual bandit problems—they are not directly suited for general RL tasks where trajectory-level preferences are crucial for solving long-horizon problems. To that end, we define a trajectory of length $n$ as $\tau = (s_0, a_0, \ldots, s_{n-1}, a_{n-1})$, and introduce the notion of a length-$k$ representative segment, denoted by $\varsigma = \Sigma(\tau, k) = (\hat{s}_0, \hat{a}_0, \ldots, \hat{s}_{k-1}, \hat{a}_{k-1})$, which approximates the overall quality and semantics of its original trajectory $\tau = \mathcal{T}(\varsigma)$. Each $(\hat{s}_t, \hat{a}_t)$ in the segment is sampled from $\tau$, with the constraint that their original indices $\mathbb{I}_\tau(t)$ are strictly increasing to preserve temporal order. We

denote a segment-level preferences as $\varsigma^+ > \varsigma^-$, which we assume it reflects overall preference for their corresponding full trajectories, i.e., $\mathcal{T}(\varsigma^+) > \mathcal{T}(\varsigma^-)$. Recent work such as Knox et al. (2024) estimates such preferences by comparing partial discounted returns $\sum_t^k \gamma^t r(s_t, a_t)$ for trajectory segments. However, in settings with sparse or highly imbalanced rewards, this return-based signal may be too weak or misleading to support reliable comparisons. To mitigate this, we instead adopt an advantage-based preference model in Contrastive Preference Learning (CPL) (Hejna et al., 2024), which focuses on distinguishing successful behaviors not just based on returns, but through their relative quality under advantage estimation:

$$P_{A^*}[\tau^+ > \tau^-] = P_{A^*}[\varsigma^+ > \varsigma^-] = \frac{\exp \sum_{\varsigma^+} \gamma^t A^*(\hat{s}_t^+, \hat{a}_t^+)}{\exp \sum_{\varsigma^+} \gamma^t A^*(\hat{s}_t^+, \hat{a}_t^+) + \exp \sum_{\varsigma^-} \gamma^t A^*(\hat{s}_t^-, \hat{a}_t^-)}, \quad (1)$$

where $A^*$ denotes the optimal advantage function, and $\tau^+ = \mathcal{T}(\varsigma^+)$ and $\tau^- = \mathcal{T}(\varsigma^-)$ are two complete trajectories. We use the shorthand "+" and "-" to denote the preferred / less preferred representative segments.

**Contrastive Preference Learning (CPL).** Hejna et al. (2024) eliminates the hidden optimal advantage function $A^*$ in the advantage-based preference model in the context of maximum entropy RL (Ziebart, 2018; Ziebart et al., 2008; Haarnoja et al., 2017). The derivation is straightforward, as Ziebart (2018) provides a critical insight, i.e., the optimal advantage function $A^*$ and optimal policy $\pi^*(a|s)$ has a direct relationship:

$$A^*(s, a) = \alpha \log \pi^*(a|s), \quad (2)$$

assuming that the optimal advantage function is normalized $\int e^{A^*(s,a)/\alpha} da = 1$. This means that instead of learning an implicit optimal advantage function, CPL can leverage the preference model to acquire the optimal policy directly. Given an offline preference dataset $\mathcal{D} = (\mathcal{D}^+, \mathcal{D}^-)$, the learning objective is to minimize the following loss function while increasing the likelihood of actions in the datasets.

$$\mathcal{L}_{\text{CPL}}(\pi_\theta, \mathcal{D}) = \mathbb{E}_{(\varsigma^+, \varsigma^-) \sim \mathcal{D}} \left[ -\log \frac{\exp \sum_{\varsigma^+} \gamma^t \alpha \log \pi_\theta(s_t^+, a_t^+)}{\exp \sum_{\varsigma^+} \gamma^t \alpha \log \pi_\theta(s_t^+, a_t^+) + \exp \lambda \sum_{\varsigma^-} \gamma^t \alpha \log \pi_\theta(s_t^-, a_t^-)} \right], \quad (3)$$

where $(\varsigma^+, \varsigma^-) \sim \mathcal{D}$ denotes drawing a pair with $\varsigma^+ \sim \mathcal{D}^+$, $\varsigma^- \sim \mathcal{D}^-$, and $\lambda \in (0, 1]$ denotes the asymmetric "bias" regularizer (An et al., 2023) that down-weights the negative segments.

## 3 PREFERENCE-BASED POLICY OPTIMIZATION

In Section 2, we reviewed contrastive preference learning (CPL) and its potential for effective policy learning in sparse-reward offline RL by leveraging a static preference dataset $\mathcal{D}_{\text{pref}}$. However, simply contrasting successful (preferred) and unsuccessful (non-preferred) behaviors does not resolve overestimation in datasets with limited state–action coverage, as the policy can still become overly optimistic in underrepresented regions due to the lack of strong counterexamples. In this section, we address the support mismatch issue by developing a practical offline RL algorithm called **PREFORL** for sparse-reward offline datasets (Qiu, 2026).

### 3.1 DEGRADATION

To mitigate the support mismatch issue, our key idea is to augment the offline dataset $\mathcal{D}$ with synthetically generated suboptimal trajectories, treated as non-preferred examples. By training policies to prefer successful trajectories over both observed and synthetic failure cases, the framework encourages imitation of high-quality behavior while simultaneously improving robustness against failure modes and distributional drift.

Specifically, given a sparse-reward offline dataset $\mathcal{D} = (\mathcal{D}^+, \mathcal{D}^-)$, we take a successful trajectory $\tau^+ \in \mathcal{D}^+$ and construct a corresponding less-preferred trajectory $\tau^-$ by applying a controlled degradation operator—either action-based ($\downarrow a$) or state-based ($\downarrow s$).

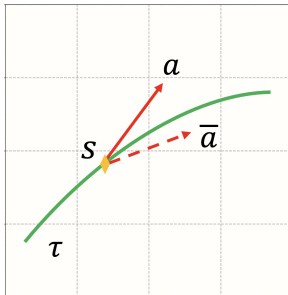 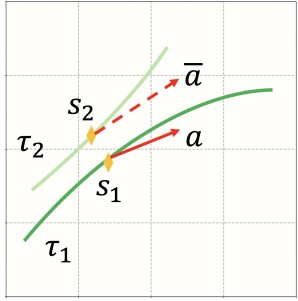

Figure 1: Two variants of degradation. In the left figure, action-based degradation ($^{\downarrow s}$) is applied to degrade the action $a$ to $\bar{a}$ using Gaussian noise. In the right figure, state-based degradation ($^{\downarrow s}$) is used to degrade the action $a$ in trajectory $\tau_1$ by finding a substitution action $\bar{a}$ correspond a neighbor state in a less preferred trajectory $\tau_2$. Red arrows with solid lines denotes the original actions, and the red arrows with dashed lines denotes degraded actions.

**Action-based Degradation** $^{\downarrow a}$. To achieve the requirement above, our action-based degradation method injects noises into the actions within the dataset (see Figure 1 left). Given $\mathcal{D}^+ = \bigcup_{i=1}^{N}\{(s_t^{(i)}, a_t^{(i)})\}_{t=1}^{T_i}$, we construct a degraded dataset $\mathcal{D}^{\downarrow a}$ by adding Gaussian noise to each action:

$$a_t^{(i)-} = a_t^{(i)} + \epsilon_t^{(i)}, \quad \epsilon_t^{(i)} \sim \mathcal{N}(0, \sigma^2 I),$$

$$\mathcal{D}^{\downarrow a} = \bigcup_{i=1}^{N}\{(s_t^{(i)}, a_t^{(i)-})\}_{t=1}^{T_i}. \tag{4}$$

Here, $\sigma$ is a tunable noise parameter whose impact is analyzed in Appendix C. By construction, the original dataset of successful trajectories is preferred over its degraded counterpart:

$$\mathcal{D}^+ > \mathcal{D}^{\downarrow a}.$$

**State-based Degradation** $^{\downarrow s}$. Instead of perturbing actions directly, state-based degradation constructs suboptimal behavior by reassigning actions from nearby states (see Figure 1 right). Given $\mathcal{D}^+ = \bigcup_{i=1}^{N}\{(s_t^{(i)}, a_t^{(i)})\}_{t=1}^{T_i}$, we build a degraded dataset $\mathcal{D}^{\downarrow s}$ by, for each state $s_t^{(i)}$, retrieving the action from a nearest-neighbor state $s_{t'}^{(j)}$ recorded in the non-preferred dataset $\mathcal{D}^-$:

$$a_t^{(i)-} = a_{t'}^{(j)}, \quad s_{t'}^{(j)} \in \texttt{NearestNeighborSearch}(\mathcal{D}^-, s_t^{(i)}),$$

$$\mathcal{D}^{\downarrow s} = \bigcup_{i=1}^{N}\{(s_t^{(i)}, a_t^{(i)-})\}_{t=1}^{T_i}. \tag{5}$$

In practice, nearest neighbors can be retrieved using Euclidean distance in vector-based state spaces or feature-space distances in high-dimensional (e.g., image-based) environments. By construction, the dataset of successful trajectories is preferred over its degraded counterpart:

$$\mathcal{D}^+ > \mathcal{D}^{\downarrow s}.$$

The philosophy behind both degradation methods is to generate suboptimal datasets that serve as the basis for constructing preference datasets for policy optimization. State-based degradation $\mathcal{D}^{\downarrow s}$ provides contrastive signals over nearby failure behaviors already present in the dataset, whereas action-based degradation $\mathcal{D}^{\downarrow a}$ introduces synthetic behaviors sampled outside the dataset's support. PREFORL can be viewed as a "squeezing" strategy: successful behaviors are sandwiched by synthetic degradations that bound what is preferable. By contrasting these degraded behaviors against successful trajectories, PREFORL trains policies not only to imitate what succeeds, but also to explicitly avoid behaviors that are likely to fail or fall outside the dataset's support.

---

**Algorithm 1** Preference-based Optimization for Offline RL (**PREFORL**)

---

**Require:** Policy parameters $\theta$, offline dataset of trajectories $\mathcal{D} = (\mathcal{D}^+, \mathcal{D}^-)$, $k$-length representative segment sampling function $\Sigma(\tau, k)$, representative segment length $k$, temperature $\alpha$, contrastive bias $\lambda$, discount factor $\gamma$.

**Ensure:** Policy $\pi_\theta(s)$

   **for** $j = 0, 1, \ldots, N-1$ **do**

      $\mathcal{D}_j^+ = \{\,\}, \mathcal{D}_j^- = \{\,\}$

      **for** $m = 0, 1, \ldots, M-1$ **do**

         $\tau_m = (\ldots, s_t, a_t, \ldots) \sim \mathcal{D}^+$

         $\varsigma = \Sigma(\tau_m, k) = (\ldots, \hat{s}_t, \hat{a}_t, \ldots, \hat{s}_l, \hat{a}_k)$            ▷ Build representative segment $\varsigma$

         $\mathcal{D}_j^+ = \mathcal{D}_j^+ \cup \{\varsigma\}$                     ▷ Collect preferred segments (+)

      **end for**

      Construct $\mathcal{D}_j^- = \mathcal{D}_j^{\downarrow a} \cup \mathcal{D}_j^{\downarrow s}$ from $\mathcal{D}_j^+$ via Equation 4 and 5

$$\theta_{j+1} = \arg\min_\theta \frac{1}{|\mathcal{D}_j^+|T} \sum_{\varsigma^\pm \in \mathcal{D}_j^\pm} \sum_{t=0}^{T-1} [-\log \frac{\exp\sum_{\varsigma^+} \gamma^t \alpha L^+}{\exp\sum_{\varsigma^+} \gamma^t \alpha L^+ + \exp\lambda \sum_{\varsigma^-} \gamma^t \alpha L^-}],$$

      where $L^\pm = \log\pi_\theta(\hat{s}_t^\pm, \hat{a}_t^\pm)$.                    ▷ Update the policy $\pi_\theta$

   **end for**

---

## 3.2 PREFORL Loss Function

Given a sparse-reward offline dataset $\mathcal{D} = (\mathcal{D}^+, \mathcal{D}^-)$, we construct a contrastive preference dataset:

$$\mathcal{D}_{\text{pref}} = (\mathcal{D}^+, \ \mathcal{D}^{\downarrow s} \cup \mathcal{D}^{\downarrow a}).$$

Unlike CPL, which contrasts $\mathcal{D}^+$ with $\mathcal{D}^-$, PREFORL instead contrasts $\mathcal{D}^+$ with the degraded datasets $\mathcal{D}^{\downarrow s} \cup \mathcal{D}^{\downarrow a}$ to guide policy learning. The PREFORL loss function $\mathcal{L}_{\text{PREFORL}}(\pi_\theta, \mathcal{D}_{\text{pref}})$ is defined as:

$$\mathbb{E}_{(\varsigma^+, \varsigma^-) \sim \mathcal{D}_{\text{pref}}} [-\log \frac{\exp\sum_{\varsigma^+} \gamma^t \alpha\log\pi_\theta(\hat{s}_t^+, \hat{a}_t^+)}{\exp\sum_{\varsigma^+} \gamma^t \alpha\log\pi_\theta(\hat{s}_t^+, \hat{a}_t^+) + \exp\lambda \sum_{\varsigma^-} \gamma^t \alpha\log\pi_\theta(\hat{s}_t^-, \hat{a}_t^-)}] \tag{6}$$

**Relation to BC and Offline RL**. PREFORL leverages preference learning to address key limitations of Behavior Cloning (BC) (Pomerleau, 1988) and offline RL. BC merely imitates demonstrations and fails under distribution shift, while offline RL often suffers from value overestimation in sparse-reward datasets with limited coverage. By contrasting successful demonstrations with synthetic degradations outside the dataset's support, PREFORL mitigates overestimation and guides policies away from brittle behaviors toward more robust and reliable trajectories. In other words, contrastive training in PREFORL encourages policies to learn not just from what succeeds, but also from what fails or lies outside the support of the dataset's distribution.

## 3.3 Algorithm Overview

We present the overview of PREFORL in Algorithm 1. Given an initial policy $\pi_\theta$, in each iteration, we sample multiple preferred representative segments $\varsigma^+$ from $\mathcal{D}^+$, and build their corresponding less preferred degraded segments $\varsigma^-$. At the end of each iteration, we optimize policy $\pi_\theta$ using PREFORL loss function shows in Equation 6. Note that PREFORL is an offline algorithm that does not requires online interaction with the environment.

## 3.4 Theoretical Justification.

Define the state marginals of $d^D$, $d^\pi$, and $d^*$ as $d^D(s)$, $d^\pi(s)$, and $d^*(s)$. Assume $\pi^*$ as the optimal policy whose induced state marginal distribution coincides with $d^*(s)$, i.e., the distribution over states visited by successful trajectories in $\mathcal{D}^+$. The following bound on the performance gap between the learned and optimal policies is established based on the above assumption in Cen et al. (2024):

$$|V^\pi(\rho_0) - V^{\pi^*}(\rho_0)| \le \frac{2R_{\max}}{1-\gamma} D_{\text{TV}}(d^*(s) \| d^D(s)) + \frac{2R_{\max}}{1-\gamma} \mathbb{E}_{d^*(s)} [D_{\text{TV}}(\pi(\cdot|s) \| \pi^*(\cdot|s))],$$

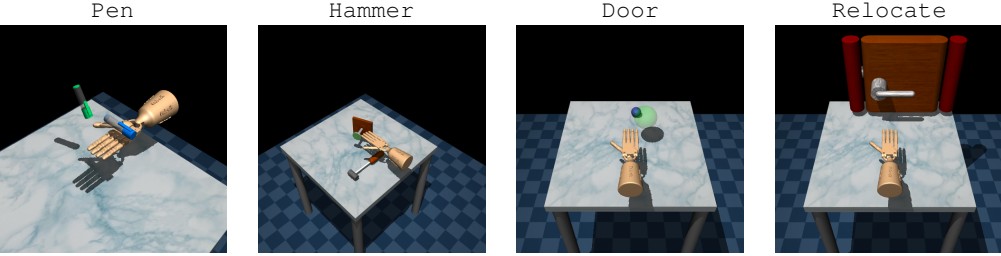

Figure 2: Hand manipulation tasks in **Adroit**.

where $R_{\max} = \max_{s,a} \|r(s,a)\|$ is the maximum reward. This shows that we can minimize $D_{\mathrm{TV}}(\pi(\cdot|s) \,\|\, \pi^*(\cdot|s))$ to optimize the learned policy $\pi$. Let $P_{A^*}(\varsigma_k^+, \varsigma_k^-) = \mathrm{Bern}\big(\frac{e^{A^*(\varsigma_k^+)}}{e^{A^*(\varsigma_k^+)} + e^{A^*(\varsigma_k^-)}}\big)$ and $P_{\hat{A}}(\varsigma_k^+, \varsigma_k^-) = \mathrm{Bern}\big(\frac{e^{\hat{A}(\varsigma_k^+)}}{e^{\hat{A}(\varsigma_k^+)} + e^{\hat{A}(\varsigma_k^-)}}\big)$, where Bern denotes Bernoulli distribution. Then the cross-entropy $\mathcal{L}_{\mathrm{PREFORL}}$ loss function in Equation 6 can be re-written in terms of the advantage functions as follows:

$$\mathcal{L}_{\mathrm{PREFORL}}(\hat{A}, \mathcal{D}_{\mathrm{pref}}) = \mathbb{E}_{(\varsigma_k^+, \varsigma_k^-) \sim \mathcal{D}_{\mathrm{pref}}} \big[ D_{KL}\big(P_{A^*}(\varsigma_k^+, \varsigma_k^-) \| P_{\hat{A}}(\varsigma_k^+, \varsigma_k^-)\big)\big]$$

We show that Algorithm 1 establishes a connection between minimizing $\mathcal{L}_{\mathrm{PREFORL}}(\hat{A}, \mathcal{D}_{\mathrm{pref}})$ and minimizing the TV divergence between the learned policy $\pi$ and the expert policy $\pi^*$.

**Lemma 3.1.** *Let* $\pi(a|s) = \frac{e^{\hat{A}(s,a)/\alpha}}{Z(s)}$ *and* $\pi^*(a|s) = \frac{e^{A^*(s,a)/\alpha}}{Z^*(s)}$, *with softmax temperature* $\alpha > 0$. *Suppose that the perturbed segments cover the full action space for each state* $s \sim d^*$. *Then:*

$$\mathcal{L}_{\mathrm{PREFORL}}(\hat{A}, \mathcal{D}_{\mathrm{pref}}) \to 0 \quad \Longrightarrow \quad \mathbb{E}_{s \sim d^*}\big[D_{\mathrm{TV}}(\pi^*(\cdot|s) \,\|\, \pi(\cdot|s))\big] \to 0.$$

The assumption that the perturbed segments cover the action space ensures that segment preferences sufficiently constrain all state-action pairs. Therefore, minimizing the loss function encourages policy imitation of the distribution of successful trajectories $\mathcal{D}^+$ in the dataset $\mathcal{D}$.

## 4 EXPERIMENTS AND EVALUATIONS

We implemented our algorithm in a tool called **PREFORL**. In this section, we evaluate PREFORL algorithm in various challenging domains including **MetaWorld** (Yu et al., 2020a), **Adroit** and **Maze2D** from D4RL (Fu et al., 2021) benchmark, and **Sparse-MuJoCo** proposed in a previous offline RL work (Cen et al., 2024).

**Adroit.** The Adroit (Rajeswaran et al., 2018) domain is designed for controlling a 24-DoF simulated Shadow Hand robot to complete different tasks. Demonstration of human experts and scripted controllers are given to evaluate the effectiveness of different RL or non-RL algorithms. In D4RL (Fu et al., 2021), Adroit is re-designed for offline RL setting only. We consider four tasks, i.e., *pen, hammer, relocate* and *door* (see Figure 2). In each task, three different types of datasets are provided to evaluate the robustness of learning algorithm. Among them, two types of datasets are adopted from the original paper (Rajeswaran et al., 2018): *human* with 25 trajectories collected from human experts, and a large amount of *expert* demonstrations sampled from a fine-tuned RL policy. Besides, each *cloned* is a mixing dataset which combines 50 percentage of expert demonstrations, and 50 percentage episodes sampled from a imitation policy trained on the demonstrations. We choose one imitation learning algorithm BC, and four offline RL algorithms (CQL, IQL, TD3+BC, CDE and ReBRAC) as baselines. Table 1 denotes the normalized scores of PREFORL algorithms against other baselines on Adroit tasks. Results indicate that PREFORL algorithm demonstrates competitive performance against other baselines and outperforms previous state-of-the-art offline RL algorithm in majority of environments.

| Task | BC | CQL | IQL | TD3+BC | CDE | ReBRAC | CPL | PREFORL |
|------|-----|-----|-----|--------|-----|--------|-----|---------|
| pen-human | 34.4 | 37.5 | 81.5±17.5 | 81.8±14.9 | 72.1 | 103.5±14.1 | 100.1±2.2 | **119.1±3.1** |
| pen-cloned | 56.9 | 39.2 | 77.2±17.7 | 61.4±19.3 | 42.1 | 91.8±21.7 | 91.2±2.2 | **92.0±3.3** |
| pen-expert | 85.1 | 107.0 | 133.6±16.0 | 146.0±7.3 | 105.0 | **154.1±5.4** | 130.9±3.2 | 144.8±3.1 |
| door-human | 0.5 | 9.9 | 3.1±2.0 | -0.1±0.0 | 7.7 | 0.0±0.0 | 11.9±0.8 | **15.5±3.2** |
| door-cloned | -0.1 | 0.4 | 0.8±1.0 | 0.1±0.6 | 0.1 | 1.1±2.6 | 3.6±3.5 | **16.3±0.7** |
| door-expert | 34.9 | 101.5 | 105.3±2.8 | 84.6±44.5 | 105.9 | 104.6±2.4 | 105.8±0.2 | 106.0±0.0 |
| hammer-human | 1.5 | 4.4 | 2.5±1.9 | 0.4±0.4 | 1.9 | 0.2±0.2 | 15.1±8.7 | **16.6±3.0** |
| hammer-cloned | 0.8 | 2.1 | 1.1±0.5 | 0.8±0.7 | 7.3 | 6.7±3.7 | 13.2±8.1 | **28.4±3.2** |
| hammer-expert | 125.6 | 86.7 | 129.6±0.5 | 117.0±30.9 | 126.3 | **133.8±0.7** | 128.3±0.3 | 128.6±0.2 |
| relocate-human | 0.0 | 0.2 | 0.1±0.1 | -0.2±0.0 | 0.3 | 0.0±0.0 | 0.6±0.0 | **0.9±0.3** |
| relocate-cloned | -0.1 | -0.1 | 0.2±0.4 | -0.1±0.1 | 0.2 | 0.9±1.6 | 0.5±0.1 | **0.9±0.1** |
| relocate-expert | 101.3 | 95.0 | 106.5±2.5 | 107.3±1.6 | 102.6 | 106.6±3.2 | 110.2±0.4 | **111.2±0.7** |

Table 1: Normalized scores of PREFORL against other baselines on D4RL **Adroit** tasks. BC, CQL and IQL scores were taken from Fu et al. (2021), TD3+BC and ReBRAC scores were taken from Tarasov et al. (2023), and CDE scores were taken from Cen et al. (2024). Our reported results are averaged over 5 random seeds, and each data point consists of 20 evaluation trajectories.

| Task | BCQ | CQL | IQL | TD3+BC | CDE | ReBRAC | CPL | PREFORL |
|------|-----|-----|-----|--------|-----|--------|-----|---------|
| halfcheetah-medium | 57.8±13.2 | 97.6±4.1 | 76.6±5.8 | 41.6±17.6 | 82.0±8.6 | **100.0** | 96.0±2.0 | 96.8±1.8 |
| walker2d-medium | 41.0±11.5 | 17.7±10.4 | 19.5±4.2 | 21.0±16.7 | 53.0±11.7 | 42.0 | 85.3±6.1 | **98.0±3.5** |
| hopper-medium | 2.0±4.0 | 74.0±5.0 | 0.0±0.0 | 0.0±0.0 | 85.5±5.7 | 96.0 | 96.0±0.0 | **100.0±0.0** |
| halfcheetah-medium-expert | 24.8±9.8 | 4.2±5.8 | 95.4±4.2 | 0.0±0.0 | 95.2±2.9 | 0.0 | 47.3±4.6 | **100.0±0.0** |
| walker2d-medium-expert | 87.0±13.4 | 61.6±23.5 | 94.6±5.9 | 32.2±22.8 | 97.0±2.8 | 36.0 | **100.0±0.0** | **100.0±0.0** |
| hopper-medium-expert | 20.0±11.0 | 0.0±0.0 | 94.8±2.8 | 22.0±10.8 | 97.0±1.4 | 21.0 | 0.0±0.0 | 98.4±3.6 |

Table 2: Success rate (in percent) of PREFORL against other baselines on **Sparase-MuJoCo**. CPL and PREFORL results are averaged over 5 random seeds, and each data point consists of 50 evaluation trajectories. Results of other baselines are taken from Cen et al. (2024) and Tarasov et al. (2023).

**Sparse-MuJoCo.** The Sparse-MuJoCo benchmark is proposed in CDE (Cen et al., 2024) and originated from MuJoCo domain in D4RL (Fu et al., 2021) benchmark. Despite all episodes are collected from inherently dense-reward based environments, the *quality* of each trajectory can be classified into *success* and *failed* categories by examining the episode return. Following the settings in CDE, the return thresholds are set to be the 75-percentile of all episode returns in each dataset. We set rewards to be 0 for all *failed* trajectories in the lower 75 percent, whereas 1 for other *success* trajectories. In evaluation, a trajectory is considered successful when the return is above the threshold and failed otherwise. On Sparse-MuJoCo, we choose BCQ, CQL, IQL, TD3+BC and CDE as baselines. These offline RL algorithms utilize different methods to optimize policies or learn value functions, and all of them leverage sparse reward information. In Table 2, PREFORL demonstrates competitive performance against other offline RL algorithms in all domains, yet only utilizing reward information *indirectly* to construct a sparse optimizing target. Details of experimental settings including dataset formulation and return thresholds can be found in Appendix H.

**Maze2D.** The Maze2D domain includes navigation tasks aiming to instruct a 2D agent to reach a fixed goal position. Three maze layouts are provided with increasing difficulties, i.e., *umaze*, *medium*, and *large* (see figures in Table 3). Different from above-mentioned domains, the training data distribution in Maze2D differs from its evaluation distribution, and the lengths of the trajectories in the dataset varies. Specifically, in data collection process, a starting position and a goal position are randomly sampled from valid positions in the maze, and an episode does not terminates if the agent reach the goal. Instead, a new goal is randomly sampled and the previous successful episode would be collected as if it is an independent trajectory. In evaluation, an agent that always start from a fixed position is required to reach as many goals as possible within maximum episode steps. These goals will be substitute by a newly randomized one if reached. Table 3 demonstrates the average returns of PREFORL and other baselines (ReBRAC, CDE) on Maze2D tasks. To enable a fair comparison with CPL on Maze2D, we introduced unsuccessful trajectories by relabeling the goal regions in a subset of the original dataset trajectories. This generates explicit failure trajectories exclusively for CPL

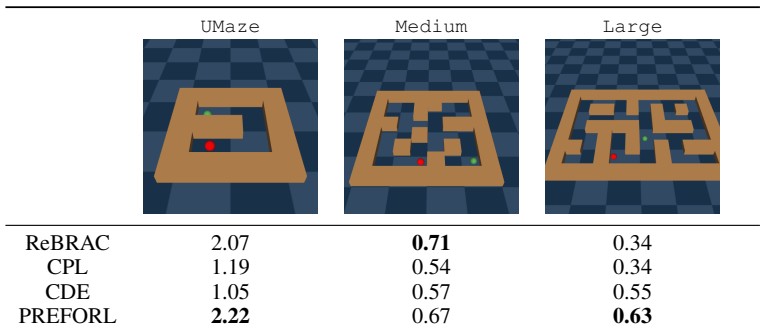

|  | UMaze | Medium | Large |
|---|---|---|---|
| ReBRAC | 2.07 | **0.71** | 0.34 |
| CPL | 1.19 | 0.54 | 0.34 |
| CDE | 1.05 | 0.57 | 0.55 |
| PREFORL | **2.22** | 0.67 | **0.63** |

Table 3: Figures show navigation tasks in different mazes in **Maze2D**. In each maze, red and green balls denote start and goal positions. The table demonstrates average numbers of successes of PREFORL against other baselines on Maze2D tasks. All results are averaged over 5 random seeds, and each data point consists of 50 evaluation trajectories. Note that we use sparse rewards in Maze2D environment, where 1 denotes one successful contact to the sampled goal and 0 otherwise.

to contrast successful versus unsuccessful rollouts. The results shows that PREFORL outperforms the baselines in most environments, and can acquire high-quality policies consistently. It also shows that PREFORL performs well in both narrow (Adroit) and diverse (Maze2D) dataset distributions. Beyond Maze2D, we additionally evaluate PREFORL on the AntMaze navigation benchmarks from D4RL (Fu et al., 2021). Detailed results are presented in Appendix G.

**MetaWorld.** The MetaWorld (Yu et al., 2020a) is a benchmark for meta-reinforcement learning and multi-task learning. It consists of 50 diverse and challenging robotic manipulation tasks. We select 16 diverse tasks from this benchmark and many of them are deemed most challenging tasks (Seo et al., 2022). Then, we use the provided scripted controller to sample 50 expert demonstrations for each selected environment. Since no reward signal is recorded, this is a typical *learning-from-demonstration* problem. To evaluate the feasibility of applying PREFORL on high-dimensional environments, we set the observation space of MetaWorld environments to be an $84 \times 84$ RGB **image**. We use BC as the sole baseline, since standard offline RL algorithms typically fail in settings where only expert demonstrations are available and reward signals are absent. To handle image-based observations, we use a pre-trained ResNet-50 (He et al., 2015) as the image encoder for both PREFORL and BC, and other training details are left in Appendix H. The evaluation results are shown in Table 4. The table shows, although BC is still a strong baseline in high-dimensional goal-achieving tasks, the PREFORL algorithm outperforms it by a large margin in nearly every domain by using sparse and limited artificial preference signals.

**Summary.** In summary, PREFORL achieves strong performance across diverse high-dimensional control tasks and consistently outperforms BC, CPL, and other strong offline RL baselines (e.g., ReBRAC, TD3+BC, CDE) in complex sparse-reward domains, demonstrating its ability to learn robust policies in the offline setting.

Our ablation studies (Appendix C) investigate the effect of the Gaussian perturbation variance $\sigma$ and find that performance remains stable within a small-noise regime. This supports our analysis that $\sigma$ must preserve locality, without requiring precise environment-specific tuning. Appendix D examines the impact of the degraded-to-preferred dataset size, showing that PREFORL is highly insensitive to the ratio and performs reliably. Appendix E provides sensitivity analyses for two key hyperparameters, the contrastive bias coefficient $\lambda$ and the representative segment length $k$, demonstrating that PREFORL maintains stable performance across a broad range of settings without requiring careful tuning. Finally, Appendix F reports additional comparisons against CPL and ReBRAC on MetaWorld under dense-reward evaluation.

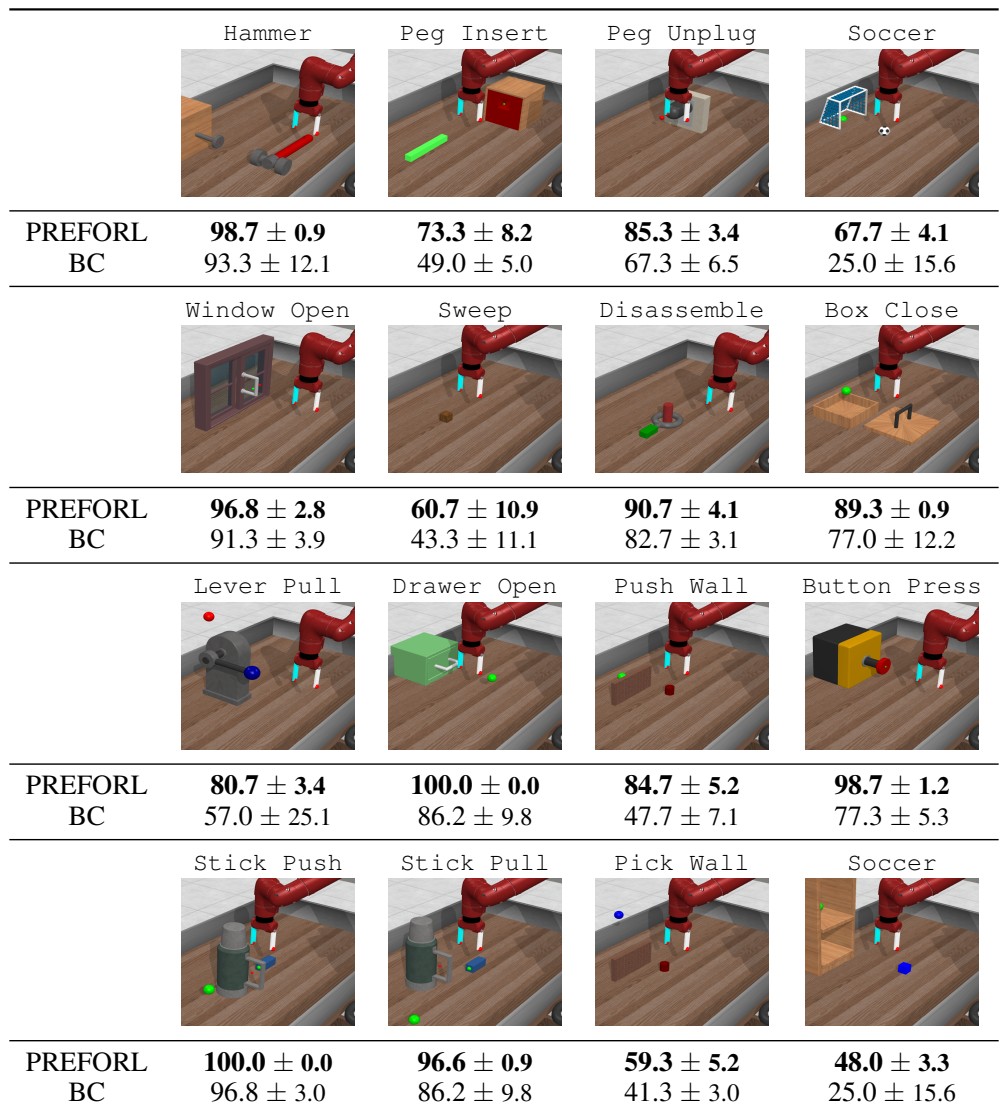

| | Hammer | Peg Insert | Peg Unplug | Soccer |
|---|---|---|---|---|
| PREFORL | **98.7 ± 0.9** | **73.3 ± 8.2** | **85.3 ± 3.4** | **67.7 ± 4.1** |
| BC | 93.3 ± 12.1 | 49.0 ± 5.0 | 67.3 ± 6.5 | 25.0 ± 15.6 |
| | Window Open | Sweep | Disassemble | Box Close |
| PREFORL | **96.8 ± 2.8** | **60.7 ± 10.9** | **90.7 ± 4.1** | **89.3 ± 0.9** |
| BC | 91.3 ± 3.9 | 43.3 ± 11.1 | 82.7 ± 3.1 | 77.0 ± 12.2 |
| | Lever Pull | Drawer Open | Push Wall | Button Press |
| PREFORL | **80.7 ± 3.4** | **100.0 ± 0.0** | **84.7 ± 5.2** | **98.7 ± 1.2** |
| BC | 57.0 ± 25.1 | 86.2 ± 9.8 | 47.7 ± 7.1 | 77.3 ± 5.3 |
| | Stick Push | Stick Pull | Pick Wall | Soccer |
| PREFORL | **100.0 ± 0.0** | **96.6 ± 0.9** | **59.3 ± 5.2** | **48.0 ± 3.3** |
| BC | 96.8 ± 3.0 | 86.2 ± 9.8 | 41.3 ± 3.0 | 25.0 ± 15.6 |

Table 4: Success rate (in percent) of PREFORL against BC on 16 tasks from **MetaWorld** benchmark. 50 expert demonstrations are provided for each environment. We report the average results of PREFORL over 5 random seeds, and each data point consists of 50 evaluation trajectories.

## 5   RELATED WORK

**Preference-based Reinforcement Learning.**    The mainstream preference-based RL (PbRL) methods often involve learning a reward model to predict the scores from pairwise comparisons, then use this reward model to perform reinforcement learning for policy optimization (Christiano et al., 2017). Early work of PbRL demonstrate the feasibility of policy learning from preference signals to solve lower-dimensional problems (Wilson et al., 2012; Akrour et al., 2012; Busa-Fekete et al., 2014), recent works, however, are able to tackle control problems by training deep neural-network policies given sufficient preference labels (Sadigh et al., 2017; Biyik & Sadigh, 2018; Ibarz et al., 2018; Shin et al., 2023; III & Sadigh, 2022). Within PbRL, Reinforcement Learning from Human Feedback (RLHF) is a special and popular paradigm that align models with human intent. By eliminating the temporal structure of RL, RLHF frame auto-regressive text-generation as a contextual bandits problem, and many algorithms (Rafailov et al., 2023; Christiano et al., 2017; Ethayarajh et al., 2024; Shao et al., 2024) proven to work well in large-scale post-training of Large Language Models in general domains (Ouyang et al., 2022; DeepSeek-AI et al., 2025a;b; Qwen et al., 2025). Parallel

to performance gains, the explainability of RLHF is also witnessing growing scholarly interest as a burgeoning research direction (Liu et al., 2025).

**Offline Reinforcement Learning.** Similar to offline RL, our work aims to optimize the policy solely from previously collected datasets without further interaction with the environment. This is particular useful in domains where online data collection is costly or unsafe. As a naive imitation learning algorithm, BC (Pomerleau, 1988) often struggles with data distributions that differ from those encountered during training. This also reveals the key challenge in offline RL: out-of-distribution (OOD) generalization. Several offline RL methods have been proposed to address the challenge. Behavior regularization approaches, such as BCQ (Fujimoto et al., 2019), BRAC (Wu et al., 2019), BEAR (Kumar et al., 2019), IQL (Kostrikov et al., 2022) and ReBRAC (Tarasov et al., 2023), restrict learned policies to stay close to the dataset's behavior distribution. Another line of work, uncertainty-aware approaches, including MOPO (Yu et al., 2020b), and CQL (Kumar et al., 2020), penalize actions with high uncertainty to mitigate the impact of distributional shift. Besides, Distribution Correction Estimation (DICE)-based methods like CDE (Cen et al., 2024) and OptiDICE (Lee et al., 2021) have been proposed to provide a direct behavior-agnostic estimation of stationary distributions to tackle offline RL problems. Recently, decision transformer (Chen et al., 2021) have introduced sequence modeling-based approaches, leveraging transformers to model trajectories directly. Approaches above optimize policies in various ways, yet they all focus on leveraging high-density transitions to construct learning objects. Our approach PREFORL, however, aggregates local experiences into a sparse preference signal, which implicitly encapsulates both step-wise knowledge for performing fine-grained control, and trajectory-level insight for discovering the optimal solution.

## 6 DISCUSSION

**Limitations.** For action-based degradation ($\downarrow^a$), variance in the Gaussian noise should be tuned accordingly. Although a sufficiently small value is well-suited in practice, knowing the action range is always preferred. For state-based degradation ($\downarrow^s$), the computation overhead is not negligible as searching valid neighbors are time-consuming. A brute-force, exact search is computationally prohibitive for large-scale datasets, which would severely limit our method's scalability. To bypass this computational bottleneck, we deliberately employ an efficient Approximated Nearest Neighbor (ANN) algorithm via FAISS library (Douze et al., 2024). This choice represents a practical trade-off between computational speed and retrieval precision. While our empirical results suggest this approximation does not harm final performance, we acknowledge that our method's effectiveness is implicitly dependent on the efficiency and quality of the ANN search algorithm.

**Conclusion.** In this work, we present a preference-based RL algorithm called PREFORL. Through extensive experiments, we demonstrate our approach can significantly outperform traditional imitation learning or offline RL algorithms in sparse-reward offline dataset. By leveraging synthetically generated negative examples, our method effectively mitigates value overestimation and learns robust policies, establishing a new state-of-the-art and a promising direction for offline policy learning. For future work, we aim to generalize PREFORL as a systematic framework for generating synthetic preference data. We plan to explore its application in LLM post-training by synthesizing negative responses to regularize the model's output distributions, thereby mitigating reward hacking without additional human labeling. Furthermore, we will explore more effective and efficient degradation and sampling strategies to automatically curate high-quality preference pairs, reducing the reliance on costly human annotations for aligning foundation models.

## REPRODUCIBILITY STATEMENT

We have included the instructions to reproduce our results in the supplementary material. The code is available on `https://github.com/RU-Automated-Reasoning-Group/PREFORL`.

## ACKNOWLEDGMENT

We thank the anonymous reviewers for their comments and suggestions. This work was supported by NSF Awards #CCF-2124155 and #CCF-2525293.

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

## THE USE OF LARGE LANGUAGE MODELS

We utilized Google's Gemini, a large language model, to assist in the writing process of this paper. Its use was strictly limited to improving grammar, clarity, and phrasing for better readability. The LLM did not contribute to any of the core research ideas, methodologies, or results presented.

## A  PROOF OF LEMMA 3.1

**Lemma 3.1** Let $\pi(a|s) = \frac{e^{\hat{A}(s,a)/\alpha}}{Z(s)}$ and $\pi^*(a|s) = \frac{e^{A^*(s,a)/\alpha}}{Z^*(s)}$, with softmax temperature $\alpha > 0$. Suppose that the perturbed segments cover the full action space for each state $s \sim d^*$. Then:

$$\mathcal{L}_{\text{PREFORL}}(\hat{A}, \mathcal{D}_{\text{PREFORL}}) \to 0 \quad \implies \quad \mathbb{E}_{s \sim d^*}\left[D_{\text{TV}}(\pi^*(\cdot|s) \,\|\, \pi(\cdot|s))\right] \to 0.$$

*Proof.* By definition,

$$\mathcal{L}_{\text{PREFORL}}(\hat{A}, \mathcal{D}_{\text{pref}}) = \mathbb{E}_{(\varsigma_k^+, \varsigma_k^-) \sim \mathcal{D}_{\text{pref}}}\left[D_{\text{KL}}\big(P_{A^*}(\varsigma_k^+, \varsigma_k^-) \,\|\, P_{\hat{A}}(\varsigma_k^+, \varsigma_k^-)\big)\right],$$

and each KL term is nonnegative. Hence the assumption

$$\mathcal{L}_{\text{PREFORL}}(\hat{A}, \mathcal{D}_{\text{pref}}) \to 0$$

implies

$$D_{\text{KL}}\big(P_{A^*}(\varsigma_k^+, \varsigma_k^-) \,\|\, P_{\hat{A}}(\varsigma_k^+, \varsigma_k^-)\big) \to 0 \quad \text{for } \mathcal{D}_{\text{pref}}\text{-a.e. segment pair.}$$

Since $P_{A^*}$ and $P_{\hat{A}}$ are Bernoulli distributions, $D_{\text{KL}}(P_{A^*} \,\|\, P_{\hat{A}}) = 0$ if and only if their success probabilities coincide (in preference learning, "success" means "prefer the positive segment"). Thus, for almost all $(\varsigma_k^+, \varsigma_k^-)$,

$$P_{A^*}(\varsigma_k^+, \varsigma_k^-) = P_{\hat{A}}(\varsigma_k^+, \varsigma_k^-).$$

Using the logistic parameterization

$$P_A(\varsigma_k^+, \varsigma_k^-) = \text{Bern}\left(\frac{e^{A(\varsigma_k^+)}}{e^{A(\varsigma_k^+)} + e^{A(\varsigma_k^-)}}\right) = \text{Bern}\big(\sigma(A(\varsigma_k^+) - A(\varsigma_k^-))\big),$$

where $\sigma$ is the sigmoid function, the equality $P_{A^*} = P_{\hat{A}}$ implies

$$\sigma\big(A^*(\varsigma_k^+) - A^*(\varsigma_k^-)\big) = \sigma\big(\hat{A}(\varsigma_k^+) - \hat{A}(\varsigma_k^-)\big).$$

Since $\sigma$ is strictly monotone, we obtain

$$A^*(\varsigma_k^+) - A^*(\varsigma_k^-) = \hat{A}(\varsigma_k^+) - \hat{A}(\varsigma_k^-) \quad \text{for } \mathcal{D}_{\text{pref}}\text{-a.e. segment pair.} \tag{7}$$

For each state $s$ in the support of $d^*$, let $(s_t, a_t)$ range over all dataset occurrences with $s_t = s$. Each such action is perturbed by Gaussian noise,

$$a_t^- = a_t + \epsilon_t, \qquad \epsilon_t \sim \mathcal{N}(0, \sigma^2 I).$$

Define

$$\mathcal{A}_s^{\text{cov}} = \{ a_t + \epsilon_t : s_t = s \}.$$

By assumption, the perturbed segments cover the full action space for each state $s \sim d^*$. Thus, $\mathcal{A}_s^{\text{cov}}$ is dense in a neighborhood of the action values relevant to $d^*$.

For a segment and its Gaussian-perturbed counterpart,

$$A^*(\varsigma_k^+) - A^*(\varsigma_k^-) = \sum_{t=0}^{k-1} \gamma^t \big(A^*(s_t, a_t) - A^*(s_t, a_t^-)\big),$$

and the same identity holds for $\hat{A}$. Define the per-step discrepancy

$$\Delta(s_t; a_t, a_t^-) = \big(\hat{A}(s_t, a_t) - \hat{A}(s_t, a_t^-)\big) - \big(A^*(s_t, a_t) - A^*(s_t, a_t^-)\big).$$

Per Equation 7, for almost all segment pairs,

$$\sum_{t=0}^{k-1} \gamma^t \, \Delta(s_t; a_t, a_t^-) = 0.$$

Because the noises $\epsilon_t$ are independent across $t$, the pairs $(a_t, a_t^-)$ arise from independent Gaussian perturbations. The identity above must hold for all such combinations. By standard uniqueness arguments for continuous functions under product Gaussian measures, each term must vanish:

$$\Delta(s_t; a_t, a_t^-) = 0 \quad \text{for almost all } (s_t, a_t, a_t^-).$$

Thus,

$$\hat{A}(s,a) - \hat{A}(s,a^-) = A^*(s,a) - A^*(s,a^-) \quad \text{for a.e. } (s,a) \text{ and } a^- \in \mathcal{A}_s^{\text{cov}}.$$

Pick any reference action $a_0 \in \mathcal{A}_s^{\text{cov}}$ and define $c(s) = \hat{A}(s,a_0) - A^*(s,a_0)$. Then on the dense set,

$$\hat{A}(s,a) = A^*(s,a) + c(s).$$

This extends to:

$$\hat{A}(s,a) = A^*(s,a) + c(s) \quad \text{for } d^*\text{-a.e. } s \text{ and } a.$$

Thus, we have:

$$
\begin{aligned}
&\mathbb{E}_{s \sim d^*} \left[ D_{\text{TV}}(\pi^*(\cdot|s) \, \| \, \pi(\cdot|s)) \right] \\
&= \mathbb{E}_{s \sim d^*} \left[ D_{\text{TV}} \left( \frac{e^{A^*(s,a)/\alpha}}{\sum_{a'} e^{A^*(s,a')/\alpha}} \, \middle\| \, \frac{e^{\hat{A}(s,a)/\alpha}}{\sum_{a'} e^{\hat{A}(s,a')/\alpha}} \right) \right] \\
&\approx \mathbb{E}_{s \sim d^*} \left[ D_{\text{TV}} \left( \frac{e^{A^*(s,a)/\alpha}}{\sum_{a'} e^{A^*(s,a')/\alpha}} \, \middle\| \, \frac{e^{(A^*(s,a)+c(s))/\alpha}}{\sum_{a'} e^{(A^*(s,a')+c(s))/\alpha}} \right) \right] \\
&\to 0.
\end{aligned}
$$

$\square$

## B  TRAINING CURVES

The training curves of full dataset experiments are shown in Figure 3. From the figure we can observe that the PREFORL converges quickly. The training curves are also stable, especially in the expert datasets training. Note that PREFORL utilizes a very sparse contrastive learning optimizing target, so that we can expect the number of gradient steps is ten times smaller than other offline RL methods (Cen et al., 2024; Tarasov et al., 2023).

## C  NOISES IN ACTION-BASED DEGRADATION

In Section 6, we discussed that for action-based degradation ($\downarrow^a$), variance $\sigma$ in the Gaussian noise should be tuned accordingly. To determine a suitable variance for PREFORL in each environment, we conduct several hyperparameter searches on the MetaWorld and Adroit tasks (in Figure 4). In all experiments, we assume we have access to the ground truth action range of each environment. This is not strictly necessary but always preferred, as approximated action ranges given by offline datasets are imprecise and conservative. The conservative estimation may lead to insufficient exploration, especially when datasets distributions are narrow.

We select two environments in MetaWorld: `Peg-Unplug` and `Peg-Insert`, and six different levels of variances. We use x% to denote variance $\sigma$ is x percentage of the environment action range. Results demonstrate that when the variance $\sigma$ is set to be a reasonably small number, i.e., around 1% to 2%, the success rates for both environments achieve their highest level. If $\sigma$ is too small (0.5%), the distinction between original actions to degraded actions may be too subtle to differentiate. This leads to insufficient exploration and optimization. Nevertheless, setting aggressive noise (e.g., 5% and 8%) may also impair the PREFORL algorithm. Applying such aggressive noises may break the semantics of the trajectories. As reflected in the proof of Lemma 3.1, PREFORL *implicitly* recovers per-state

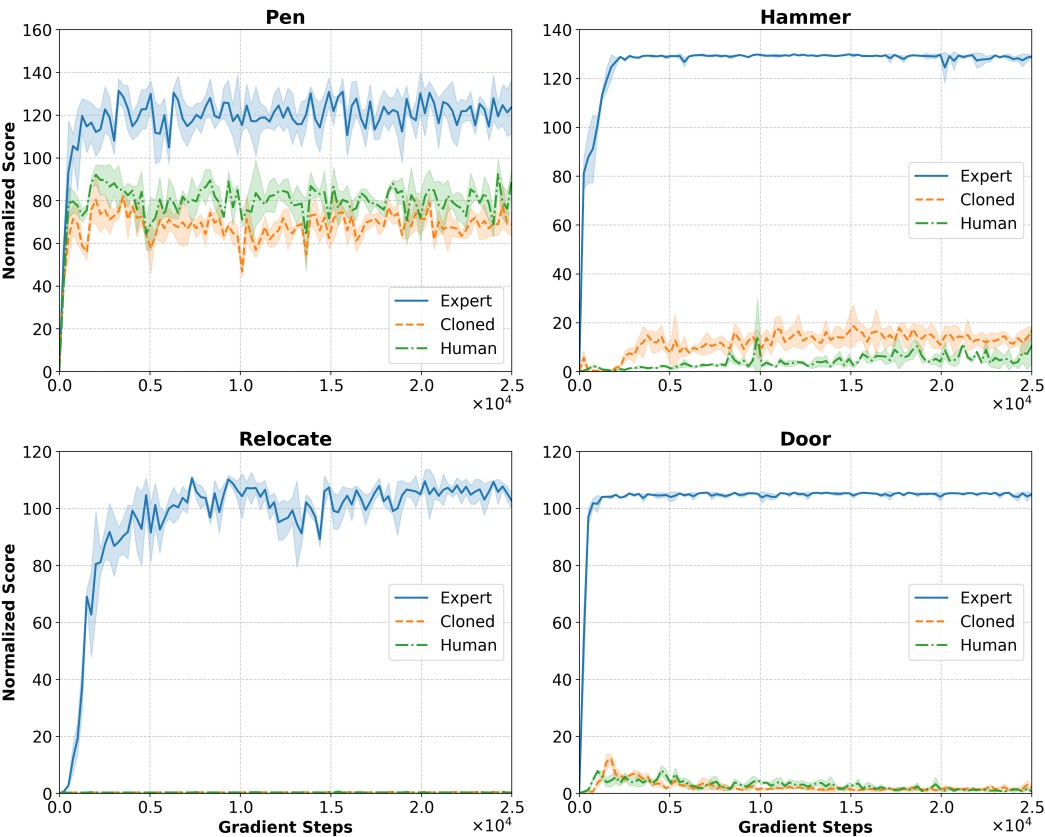

Figure 3: The training curves of PREFORL. The x-axis denotes number of gradient steps and the y-axis denotes the normalized scores. We use 5 random seeds for each environment, and each data point consists of 50 evaluation trajectories. The shadow regions denote the standard deviation of mean values across different seeds.

advantage differences from segment-level preferences, which requires action perturbations to be local so that the differences $A^*(s, a) - A^*(s, a')$ capture the local geometry of $A^*(s, \cdot)$; large-variance perturbations violate this assumption and break this identifiability step. The lemma clarifies why the noise variance $\sigma$ must be reasonably small in order to satisfy the locality requirement in our analysis.

We select all four expert datasets in Adroit: *Pen-Expert, Door-Expert, Hammer-Expert* and *Relocate-Expert*, and evaluate the effect of four different "small" variances accordingly. In Figure 4, the success rates for Adroit tasks keep unchanged across different levels of Gaussian noises. The result shows that the success rates are stable across a range of Gaussian noise levels, as long as $\sigma$ remains in the small-noise regime. This indicates that the perturbation variance does not require significant environment-specific tuning.

# D ABLATION STUDY OF DEGRADED DATASET SIZE

We performed an ablation study varying the size of the degraded dataset by changing the degraded-to-preferred ratio ($|\mathcal{D}^{\downarrow s} \cup \mathcal{D}^{\downarrow a}|/|\mathcal{D}^+|$) from 1 to 10, and also considered the limiting case where degraded trajectories are generated on-the-fly ($\infty$) during training loops. As shown in Table 5, performance remains stable across all settings: the normalized scores vary by less than 1–2 points on every Adroit task, and no consistent trend emerges as the ratio increases. This indicates that PREFORL is insensitive to the size of degraded samples and does not require a large or exhaustively constructed degraded dataset.

This empirical robustness is consistent with our theory. The theoretical condition in Lemma 3.1 requires that the noise distribution has full support over a local neighborhood in the action space—not

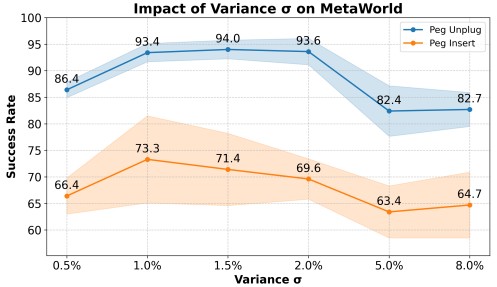 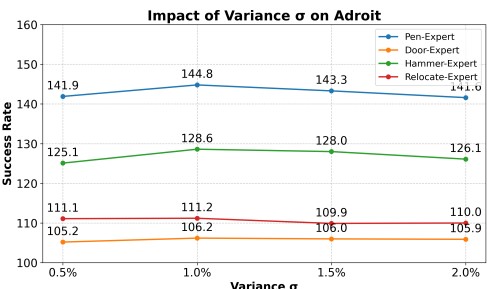

Figure 4: The left figure denotes the effect of noise level on two MetaWorld tasks. The right figure denotes the effect of noise level on Adroit tasks with expert dataset. Results are averaged over 5 run seeds, and each data point is collect by 50 evaluation trajectories.

| Task | Ratio = $\infty$ | Ratio = 1 | Ratio = 5 | Ratio = 10 |
|---|---|---|---|---|
| door-expert | 106.0 | 106.1 | 105.2 | 105.9 |
| hammer-expert | 128.6 | 128.3 | 128.4 | 128.3 |
| pen-expert | 144.8 | 143.1 | 144.0 | 143.3 |
| relocate-expert | 111.1 | 111.0 | 109.3 | 111.1 |

Table 5: Impact of the degraded-to-preferred sample ratio on Adroit tasks. Scores are normalized and averaged over 3 seeds; $\infty$ denotes on-the-fly degraded sample generation.

that the dataset enumerates this support exhaustively. Gaussian perturbation already satisfies this condition. In practice, drawing a modest number of perturbed trajectories provides sufficient "probabilistic coverage" to estimate the expectation in the preference loss.

## E  SENSITIVITY ANALYSIS

We include sensitivity analyses for both the contrastive bias parameter $\lambda$ and the representative segment length $k$, and the results shown in below tables are included in Tables 6 and 7 of the revised manuscript.

As shown in Table 6, PREFORL is robust to the choice of $\lambda$ over a wide range of values (0.25, 0.5, 1.0). Performance varies only slightly across settings and remains consistently strong on all Adroit tasks. Our default choice is $\lambda = 0.5$.

Similarly, Table 7 demonstrates that PREFORL is insensitive to the segment length $k$. Across $k \in \{20, 50, 100, 150\}$ (with total sequence length fixed at 200), performance on pen-human and relocate-expert remains stable, with only minor fluctuations. Our default setting is $k = 100$. Overall, these ablations show that PREFORL does not require fine-tuning of either $\lambda$ or $k$; the method is stable across a broad range of hyperparameter choices.

## F  COMPARSION AGAINST CPL AND REBRAC ON METAWORLD

We include comparisons agains CPL and ReBRAC on MetaWorld tasks in Table 8. For each task, the dataset contains 50 demonstrations. We intentionally augmented the demonstration dataset with environment reward information to give ReBRAC a stronger supervisory signal, whereas PREFORL does not rely on any dense reward annotations.

Both CPL and ReBRAC struggle on the demonstration-only MetaWorld tasks we evaluated, whereas PREFORL consistently achieves 90–100% success. On disassemble-v2 and stick-push-v2, both CPL and ReBRAC fail completely, while PREFORL reaches 90.7% and 100%, respectively. On door-open-v2 and hammer-v2, where CPL or ReBRAC show stronger performance, PREFORL still matches or exceeds them, achieving 100% and 98.7%. These results suggest that, at least in the

| Task | $\lambda = 0.25$ | $\lambda = 0.5$ | $\lambda = 1$ |
|---|---|---|---|
| pen-human | $113.9 \pm 2.4$ | $119.1 \pm 3.1$ | $116.4 \pm 3.2$ |
| relocate-expert | $110.7 \pm 0.1$ | $112.2 \pm 0.7$ | $110.5 \pm 0.3$ |
| door-expert | $105.7 \pm 0.1$ | $106.0 \pm 0.0$ | $105.5 \pm 0.3$ |
| hammer-expert | $127.7 \pm 0.4$ | $128.6 \pm 0.2$ | $122.5 \pm 0.5$ |

Table 6: Ablation study on contrastive bias $\lambda$ for Adroit tasks.

| Segment Length ($k$) | pen-human | relocate-expert |
|---|---|---|
| 20 | $119.0 \pm 0.8$ | $110.6 \pm 1.1$ |
| 50 | $118.7 \pm 3.9$ | $110.6 \pm 1.1$ |
| 100 | $119.1 \pm 3.1$ | $112.2 \pm 0.7$ |
| 150 | $121.1 \pm 5.19$ | $111.7 \pm 0.9$ |

Table 7: Ablation study on segment length $k$ for Adroit tasks (with total sequence length fixed at 200).

subset of MetaWorld environments we examined, preference-based learning with locally degraded actions provides a more reliable supervision signal than methods that rely on inferring value functions from sparse or heterogeneous demonstrations. In the failure cases for ReBRAC, we found that higher learned reward estimates or critic values do not necessarily correlate with higher task success.

## G  EVALUATION ON ANTMAZE ENVIRONMENTS

We selected a subset of the AntMaze suite (umaze-diverse-v2, large-play-v2, and large-diverse-v2) where existing offline RL baselines, particularly IQL and ReBRAC, are less effective according to their results. The results are reported in Table 9.

PREFORL achieves competitive performance on two of the three difficult tasks. In particular, large-play-v2 is the setting where preference-based learning provides the clearest benefit: demonstrations in "play" datasets are noisy and locally inconsistent. PREFORL relies on local preference signals induced by small action degradations rather than global return estimation, allowing it to extract reliable supervision even when the offline data are highly suboptimal. On the other hand, PREFORL is less effective on large-diverse-v2. This dataset contains highly heterogeneous demonstrations that traverse many disconnected regions of the maze with less state overlap. As a result, the local preference pairs used by PREFORL become sparse in the parts of the state space relevant for reaching the goal, making it difficult for the model to propagate local improvements across the entire maze. In contrast, ReBRAC performs well on this task because it leverages global value-function regularization that benefits from the broad coverage in the diverse dataset. We believe this result provides useful insight into the strengths and limitations of preference-based offline RL.

## H  EXPERIMENT DETAILS

We use $8 \times$ A100 80G Nvidia GPUs for experiments. In this section, we discuss more experiment details including task formulation and dataset construction, as well as training details including network architectures and hyperparameters.

### H.1  ADROIT

**Tasks.** The Adroit in D4RL (Fu et al., 2021) contains four manipulation tasks (*pen, hammer, door* and *relocate*), and three types of datasets (*expert, human* and *cloned*). In human setting, 25 human-generated high-quality trajectories are collected in each dataset. In expert datasets, a scripted controller is used to generate 5K successful trajectories for generating offline dataset. However, cloned is a special dataset that contains 5K both success and failed trajectories, as half of the episodes are collected from expert demonstrations, and the other half are sampled from an suboptimal

| Task Name | CPL | ReBRAC | PREFORL |
|---|---|---|---|
| disassemble-v2 | 0.0 | 0.0 | 90.7 |
| stick-push-v2 | 0.0 | 0.0 | 100.0 |
| door-open-v2 | 100.0 | 84.4 | 100 |
| hammer-v2 | 100.0 | 0.0 | 98.7 |

Table 8: Comparison of CPL, ReBRAC, and PREFORL on Meta-World Tasks.

| Dataset | BC | TD3+BC | AWAC | CQL | IQL | ReBRAC | DT | PREFORL |
|---|---|---|---|---|---|---|---|---|
| umaze-diverse-v2 | 47.3±4.1 | 44.8±11.6 | 54.8±8.0 | 37.3±3.7 | 54.3±5.5 | **83.5±7.0** | 51.8±0.4 | 68.9±3.2 |
| large-play-v2 | 0.0±0.0 | 0.0±0.0 | 0.0±0.0 | 20.8±7.3 | 42.0±4.5 | 52.3±29.0 | 0.0±0.0 | **53.2±14.2** |
| large-diverse-v2 | 0.0±0.0 | 0.0±0.0 | 0.0±0.0 | 20.5±13.2 | 30.3±3.6 | **64.0±5.4** | 0.0±0.0 | 26.1±7.1 |

Table 9: Average scores on selected D4RL **AntMaze** tasks. PREFORL results are averaged over 3 seeds.

imitation policy. The Adroit dataset is a typical narrow distribution dataset because the tasks in Adroit contains relatively fixed goals and traces. This makes it suitable for both action-based and state-based degradation. In practice, we use Approximated Nearest Neighbor (ANN) search method `IndexIVFFlat` implemented in FAISS (Douze et al., 2024) to search 10 nearest neighbor states. If any state with 10 percent less reward is found, we use its corresponding action as the degraded action to perform contrastive learning.

**Hyperparameters.** The hyperparameters listed in Table 10 are used to train Adroit policies using PREFORL algorithm.

## H.2 METAWORLD

**Tasks.** We set tasks in MetaWorld with image-based observations; hence, we need a network structure to process the RGB image. We choose a pre-trained ResNet-50 (He et al., 2015) model as the image encoder for both BC and PREFORL[1]. Unlike many previous works, we do not freeze the ResNet model during training.

**Hyperparameters.** The hyperparameters listed in Table 11 are used to train MetaWorld policies using PREFORL algorithm.

## H.3 MAZE2D

**Hyperparameters.** The hyperparameters listed in Table 12 are used to train Maze2D policies using PREFORL algorithm.

## H.4 SPARSE-MUJOCO

**Tasks.** We adopt "-v2" tasks in D4RL for Sparse-MuJoCo domains. We convert dense rewards in the offline dataset into sparse rewards as stated in the main text, and return thresholds for each environment are listed in Table 13. Note that we perform binary judgment in evaluation, i.e., a trajectory is considered successful if, and only if, its return exceeds the corresponding threshold.

**Hyperparameters.** The hyperparameters listed in Table 14 are used to train Sparse-MuJoCo policies using PREFORL algorithm.

---

[1]Model is available at: https://download.pytorch.org/models/resnet50-11ad3fa6.pth

| Description | Value |
|---|---|
| Discount factor $\gamma$ | 1.0 |
| Biased regularizer value $\alpha$ | 0.1 |
| Contrastive bias $\lambda$ | 0.5 |
| Contrastive segments length $l$ | 64 |
| Batch size | 20 |
| Number of gradient steps | 15000 |
| Learning rate | 0.0003 |
| Degradation operators | Action-based $\downarrow a$, State-based $\downarrow s$ |
| Variance $\sigma$ in action-based $\downarrow a$ | 1% |
| Nearest neighbor search in state-based $\downarrow s$ | `IndexIVFFlat` |
| Number of probes | 10 |
| Condition `cond` in state-based $\downarrow s$ | Reward is at least 10% smaller |
| Policy network | MLP (1024, 1024, 1024) |
| Activation | ReLU |

Table 10: Hyperparameters of PREFORL in training **Adroit** tasks.

| Description | Value |
|---|---|
| Discount factor $\gamma$ | 1.0 |
| Biased regularizer value $\alpha$ | 0.1 |
| Contrastive bias $\lambda$ | 0.5 |
| Contrastive segments length $l$ | 100 |
| Batch size | 64 |
| Number of gradient steps | 15000 |
| Learning rate | 0.0003 |
| Degradation operators | Action-based $\downarrow a$ |
| Variance $\sigma$ in action-based $\downarrow a$ | 1% |
| Image encoder | Pre-trained `ResNet-50`[2] |
| Policy network | MLP (1024, 1024, 1024) |
| Activation | ReLU |

Table 11: Hyperparameters of PREFORL in training **MetaWorld** tasks.

| Description | Value |
|---|---|
| Discount factor $\gamma$ | 1.0 |
| Biased regularizer value $\alpha$ | 0.1 |
| Contrastive bias $\lambda$ | 0.5 |
| Contrastive segments length $l$ | 100 |
| Batch size | 64 |
| Number of gradient steps | 500 |
| Learning rate | 0.0003 |
| Degradation operators | Action-based $\downarrow a$ |
| Variance $\sigma$ in action-based $\downarrow a$ | 1% |
| Policy network | MLP (1024, 1024, 1024) |
| Activation | ReLU |

Table 12: Hyperparameters of PREFORL in training **Maze2D** tasks.

| Task | Return Threshold |
|---|---|
| halfcheetah-medium | 4909.1 |
| walker2d-medium | 3697.8 |
| hopper-medium | 1621.5 |
| halfcheetah-medium-expert | 10703.4 |
| walker2d-medium-expert | 4924.8 |
| hopper-medium-expert | 3561.9 |

Table 13: The return thresholds for **Sparse-MuJoCo** tasks.

| Description | Value |
|---|---|
| Discount factor $\gamma$ | 1.0 |
| Biased regularizer value $\alpha$ | 0.1 |
| Contrastive bias $\lambda$ | 0.5 |
| Contrastive segments length $l$ | 100 |
| Batch size | 64 |
| Number of gradient steps | 15000 |
| Learning rate | 0.0003 |
| Degradation operator | Action-based $^{\downarrow a}$ |
| Variance $\sigma$ in action-based $^{\downarrow a}$ | 1% |
| Policy network | MLP (1024, 1024, 1024) |
| Activation | ReLU |

Table 14: Hyperparameters of PREFORL in training **Sparse-MuJoCo** tasks.

