# OpenReview forum: "Preference-based Policy Optimization from Sparse-reward Offline Dataset"
_ICLR.cc/2026/Conference — ICLR 2026 Poster_

### Official Review · Reviewer_uvFa · 2025-10-25

**Soundness:** 2
**Presentation:** 3
**Contribution:** 3
**Rating:** 6
**Confidence:** 3

**Summary:**

This work presents a preference-based RL algorithm, which trains policies by contrasting successful demonstrations with failure behaviors present in the dataset. Experiments on offline benchmarks with sparse rewards validate the effectiveness of the proposed method.

**Strengths:**

- This paper proposes a contrastive preference learning framework to bypass direct value function estimation.
- This paper provides both empirical and theoretical analyses.
- The proposed approach outperforms baselines in various benchmarks.

**Weaknesses:**

- The motivation is not adequately supported by evidence. The authors claim that existing methods are sensitive to support mismatches and prone to high variance or instability, particularly when data are limited or rewards are sparse, but no empirical results or references are provided to support these statements.
- The paper does not provide results with other competitive baselines on MetaWorld, such as PREFORL [1] and CPL [2].
- There is no sensitivity analysis on the representative segment length $k$ and the contrastive bias $\lambda$.

References:

[1] Tarasov et al. "Revisiting the Minimalist Approach to Offline Reinforcement Learning", NeurIPS, 2023.

[2] Hejna et al. "Contrastive Preference Learning: Learning from Human Feedback without RL", ICLR, 2024.

**Questions:**

How is MetaWorld configured to use sparse rewards? By default, MetaWorld provides dense reward settings.

---

> ### Author Response · Authors · 2025-11-26
>
> We appreciate your insightful feedback and constructive comments!
>
> **The motivation is not adequately supported by evidence. The authors claim that existing methods are sensitive to support mismatches and prone to high variance or instability, particularly when data are limited or rewards are sparse, but no empirical results or references are provided to support these statements.**
>
> We have revised the introduction section to include concrete citations supporting the statement that existing offline RL methods are sensitive to distributional shift and can exhibit overestimation, high variance, or instability, especially under sparse rewards or limited data. Regarding empirical evidence, our results in Tables 1–3 reflect these phenomena. Standard offline RL baselines show substantial performance degradation, high variance across seeds, and instability across tasks, consistent with the challenges reported in prior work and illustrating the precise failure modes we refer to.
>
> **There is no sensitivity analysis on the representative segment length $k$ and the contrastive bias $\lambda$.**
>
> We have added sensitivity analyses for both the contrastive bias parameter $\lambda$ and the representative segment length $k$, and the results shown in below tables are included in Tables 6 and 7 in Appendix E of the revised manuscript.
>
> PREFORL is robust to the choice of $\lambda$ over a wide range of values (0.25, 0.5, 1.0). Performance varies only slightly across settings and remains consistently strong on all Adroit tasks. Our default choice is $\lambda=0.5$.
>
> **Table:** Ablation study on contrastive bias λ for Adroit tasks.
> | Task            | λ = 0.25        | λ = 0.5          | λ = 1            |
> |-----------------|------------------|-------------------|-------------------|
> | pen-human       | 113.9 ± 2.4      | 119.1 ± 3.1   | 116.4 ± 3.2       |
> | relocate-expert | 110.7 ± 0.1      | 112.2 ± 0.7   | 110.5 ± 0.3       |
> | door-expert     | 105.7 ± 0.1      | 106.0 ± 0.0   | 105.5 ± 0.3       |
> | hammer-expert   | 127.7 ± 0.4      | 128.6 ± 0.2   | 122.5 ± 0.5       |
>
> Similarly, PREFORL is insensitive to the segment length $k$. Across $k \in$ {20, 50, 100, 150} (with total sequence length fixed at 200), performance on pen-human and relocate-expert remains stable, with only minor fluctuations. Our default setting is $k=100$. Overall, these ablations show that PREFORL does not require fine-tuning of either $\lambda$ or $k$; the method is stable across a broad range of hyperparameter choices.
>
> **Table:** Ablation study on segment length *k* for Adroit tasks (total sequence length fixed at 200).
> | Segment Length (k) | pen-human        | relocate-expert   |
> |--------------------|------------------|--------------------|
> | 20                 | 119.0 ± 0.8      | 110.6 ± 1.1        |
> | 50                 | 118.7 ± 3.9      | 110.6 ± 1.1        |
> | 100                | 119.1 ± 3.1      | 112.2 ± 0.7    |
> | 150                | 121.1 ± 5.19 | 111.7 ± 0.9        |
>
> **How is MetaWorld configured to use sparse rewards? By default, MetaWorld provides dense reward settings.**
>
> MetaWorld indeed provides dense reward functions by default. In our setting, however, we do not use the environment’s reward signal. We simply use the provided scripted controller to collect 50 expert demonstrations for each selected task, and no reward values are recorded. As a result, our experiments operate purely in a learning-from-demonstration setting on MetaWorld, where the algorithm relies only on trajectory data rather than environment rewards.
>
> **The paper does not provide results with other competitive baselines on MetaWorld, such as ReBRAC and CPL.**
>
> We have incorporated comparisons against CPL and ReBRAC. The results appear in Table 8 of Appendix F in the revised paper (reproduced below). For each task, the dataset contains 50 demonstrations. We intentionally augmented the demonstration dataset with environment reward information to give ReBRAC a stronger supervisory signal, whereas PREFORL does not rely on any dense reward annotations.
>
> **Table:** Comparison of CPL, ReBRAC, and PREFORL Success Rates on MetaWorld.
> | Tasks        | CPL   | ReBRAC | PREFORL |
> |------------------|-------|--------|---------|
> | disassemble-v2   | 0.0   | 0.0    | 90.7    |
> | stick-push-v2    | 0.0   | 0.0    | 100.0   |
> | door-open-v2     | 100.0 | 84.4   | 100.0   |
> | hammer-v2        | 100.0 | 0.0    | 98.7    |
>
> Both CPL and ReBRAC struggle on some MetaWorld tasks we evaluated, whereas PREFORL consistently achieves 90–100\% success. In the failure cases for ReBRAC, we found that higher learned reward estimates or critic values do not necessarily correlate with higher task success. These results suggest that (at least in the subset of MetaWorld environments we examined) preference-based learning with locally degraded actions provides a more reliable supervision signal than methods that rely on inferring value functions from sparse or heterogeneous demonstrations.

---

> > ### Comment · Reviewer_uvFa · 2025-11-26
> >
> > Thanks for your responses. I will keep my rating.

---

### Official Review · Reviewer_k6nG · 2025-10-28

**Soundness:** 2
**Presentation:** 3
**Contribution:** 2
**Rating:** 6
**Confidence:** 3

**Summary:**

This paper proposes a novel method named PREFORL, which utilizes a contrastive learning framework to learn preferences from successful trajectories and synthetically degraded trajectories, aiming to address the value overestimation problem in sparse-reward offline reinforcement learning.

**Strengths:**

1) The idea is novel, introducing the concept of preference learning into offline RL to mitigate overestimation.

2) It designs a scheme for generating negative trajectories and validates the algorithm's effectiveness through extensive experiments.

**Weaknesses:**

1) The underlying theory and mechanism explaining why introducing the preference learning framework alleviates overestimation are unclear.

2) The method for generating negative trajectories and the selection of the contrastive bias parameter vary across different tasks, and their impact on performance remains unknown.

3) The proof for Lemma 3.1 is not rigorous, as the approximation between $\hat{A}$ and $A^*$ is unreasonable.

**Questions:**

Please see weaknesses.

---

> ### Author Response · Authors · 2025-11-26
>
> We appreciate your insightful feedback and constructive comments!
>
> **The underlying theory and mechanism explaining why introducing the preference learning framework alleviates overestimation are unclear.**
>
> In classical offline RL, overestimation is typically caused by evaluating or extrapolating $Q$-values on out-of-distribution (OOD) actions. Our method avoids this failure mode by never learning absolute value estimates; instead, it learns *relative preferences* induced by local action perturbations. Intuitively, each preference constraint compares a successful segment $\varsigma^{+}$ with a perturbed segment $\varsigma^{-}$ that visits the same states but takes slightly altered actions. These constraints enforce the correct local ordering of action quality at each state. This means the learned policy cannot assign higher probability to suboptimal or off-manifold action without violating preference constraints. In other words, PREFORL can be viewed as a *squeezing* mechanism: successful behaviors are "sandwiched" between synthetic degradations that provide lower bounds on what is preferable. By contrasting these degraded behaviors against successful trajectories, PREFORL trains the policy not only to imitate behaviors that succeed, but also to explicitly avoid behaviors that lie outside the support of the dataset. Our explanation for why PREFORL mitigates overestimation aligns with the theoretical guarantee in Lemma 3.1. The lemma states that, under local Gaussian perturbations of the actions, driving the preference loss to zero forces the learned policy to match the optimal policy on the optimal state distribution $d^{\ast}$, i.e., $\mathbb{E}{s \sim d^{\ast}}[D_{\mathrm{TV}}(\pi^{\ast}(\cdot \mid s)\ |\ \pi(\cdot \mid s))] \to 0$.
>
> **The method for generating negative trajectories and the selection of the contrastive bias parameter vary across different tasks, and their impact on performance remains unknown.**
>
> We apologize for the confusion—our presentation may have obscured what is shared versus task-specific. Conceptually, PREFORL uses a single preference-construction mechanism (Algorithm 1) across all benchmarks. While the source of successful/failed trajectories ($\mathcal{D}^+$/$\mathcal{D}^-$) naturally depends on the supervision available in each benchmark (expert demos or thresholded returns), the degraded trajectory construction used by PREFORL itself to derive state-based and action-based degradation $\mathcal{D}^{\downarrow s} \cup \mathcal{D}^{\downarrow a}$ remains consistent across all tasks. Formally, given any sparse-reward offline dataset $\mathcal{D} = (\mathcal{D}^+, \mathcal{D}^-)$,
> PREFORL automatically constructs a contrastive preference dataset:
> $
> \mathcal{D}_{\texttt{pref}} = \bigl(\mathcal{D}^+,
> \mathcal{D}^{\downarrow s} \cup \mathcal{D}^{\downarrow a}\bigr)
> $ for preference-based policy optimization.
>
> We have added sensitivity analyses for the contrastive bias parameter $\lambda$ and the results shown in the following table are included as Table 6 in Appendix E in the revised manuscript.
>
> **Table:** Ablation study on contrastive bias λ for Adroit tasks.
> | Task            | λ = 0.25        | λ = 0.5          | λ = 1            |
> |-----------------|------------------|-------------------|-------------------|
> | pen-human       | 113.9 ± 2.4      | 119.1 ± 3.1   | 116.4 ± 3.2       |
> | relocate-expert | 110.7 ± 0.1      | 112.2 ± 0.7   | 110.5 ± 0.3       |
> | door-expert     | 105.7 ± 0.1      | 106.0 ± 0.0   | 105.5 ± 0.3       |
> | hammer-expert   | 127.7 ± 0.4      | 128.6 ± 0.2   | 122.5 ± 0.5       |
>
> PREFORL is robust to the choice of $\lambda$ over a wide range of values (0.25, 0.5, 1.0). Performance varies only slightly across settings and remains consistently strong on all Adroit tasks. Our default choice is $\lambda=0.5$.
>
> **The proof for Lemma 3.1 is not rigorous.**
>
> We thank the reviewer for pointing out that the original proof of Lemma 3.1 was not fully rigorous and relied on an informal “approximation” between $\hat{A}$ and $A^\ast$. We have substantially revised the proof to address this concern in Appendix A.

---

### Official Review · Reviewer_aGGo · 2025-11-01

**Soundness:** 3
**Presentation:** 3
**Contribution:** 2
**Rating:** 6
**Confidence:** 2

**Summary:**

This paper presents PREFORL (PREFerence-based Optimization for Offline RL), a novel contrastive preference learning framework designed to train robust policies from sparse-reward offline datasets. The method aims to bypass the core challenges of conventional offline RL, specifically the extrapolation error and overestimation bias that plague value-based methods in data-limited, sparse-reward settings.

**Strengths:**

1.  The key idea is fundamentally original: avoiding direct, unstable value function estimation (the common bottleneck in sparse-reward offline RL) by transforming the task into a more robust contrastive learning problem.

2. The introduction of two controlled degradation operators ($\mathcal{D}^{\perp a}$ and $\mathcal{D}^{\perp s}$) is a highly creative mechanism for generating meaningful synthetic negative examples.

**Weaknesses:**

1. For state-based degradation ($\mathcal{D}^{\perp s}$), the computation overhead associated with Nearest Neighbor Search is acknowledged to be non-negligible and potentially time-consuming for large-scale datasets. A brute-force, exact search is computationally prohibitive.

2. The performance of the action-based degradation method is highly sensitive to the choice of the noise variance ($\sigma$), which limits its robustness. This requires manual tuning per environment (or environment domain) to find the reasonably small number (e.g., $1\%$ to $2\%$) that maximizes the success rate.

**Questions:**

The CPL baseline is excluded from the Maze2D evaluation, with the justification that the dataset lacks unsuccessful trajectories. Given that PREFORL's core novelty is to augment these negative examples, can you make a direct comparison showing that CPL fails while PREFORL succeeds due to the synthetic degradation? It would have been a more powerful demonstration.

---

> ### Author Response · Authors · 2025-11-26
>
> We appreciate your insightful feedback and constructive comments!
>
> **For state-based degradation, the computation overhead associated with Nearest Neighbor Search is acknowledged to be non-negligible and potentially time-consuming for large-scale datasets. A brute-force, exact search is computationally prohibitive.**
>
> For state-based degradation, we agree that a brute-force, exact nearest neighbor search can be expensive for very large datasets. In our experiments, this degradation step is performed *once offline with approximation*, so the overhead is small compared to the cost of training the policy itself. More broadly, our method only requires a mechanism to generate locally lower-quality actions around the states in $\mathcal{D}^+$; nearest-neighbor reuse from $\mathcal{D}^-$ is just one simple instantiation. In larger-scale settings, one could use a learned conditional model over $\mathcal{D}^-$. For example, one can train a conditional behavior model $\pi^{-}\_\phi(a \mid s)$ (e.g., via behavior cloning or density estimation) on $\mathcal{D}^-$, and then construct the degraded dataset by sampling $a_t^{(i)-} \sim \pi^{-}_\phi(\cdot \mid s_t^{(i)})$ for state $s_t^{(i)} \in \mathcal{D}^+$. This amortizes the cost of degradation and avoids explicit nearest-neighbor search, while still producing locally suboptimal actions grounded in the empirical failure data. Exploring such scalable instantiations is orthogonal to our main contribution and we view them as natural extensions of the framework rather than limitations of the approach.
>
> **The performance of the action-based degradation method is highly sensitive to the choice of the noise variance $\sigma$, which limits its robustness. This requires manual tuning per environment (or environment domain) to find the reasonably small number that maximizes the success rate.**
>
> As reflected in the proof of Lemma 3.1, PREFORL *implicitly* recovers per-state advantage differences from segment-level preferences, which requires action perturbations to be local so that the differences $A^\ast(s,a)-A^\ast(s,a^-)$ capture the local geometry of $A^\ast(s,\cdot)$; large-variance perturbations violate this assumption and break this identifiability step. The theorem clarifies why the noise variance $\sigma$ must be reasonably small in order to satisfy the locality requirement in our analysis.
>
> Empirically, Figure 4 in the appendix shows that the Adroit success rates are stable across a range of Gaussian noise levels, as long as $\sigma$ remains in the small-noise regime. This indicates that the perturbation variance $\sigma$ does not require significant environment-specific tuning.
>
> **The CPL baseline is excluded from the Maze2D evaluation, with the justification that the dataset lacks unsuccessful trajectories. Given that PREFORL's core novelty is to augment these negative examples, can you make a direct comparison showing that CPL fails while PREFORL succeeds due to the synthetic degradation? It would have been a more powerful demonstration.**
>
> We thank the reviewer for the great suggestion. To enable a fair comparison with CPL on Maze2D, we introduced unsuccessful trajectories by relabeling the goal regions in a subset of the original dataset trajectories. This generates explicit failure trajectories that CPL needs in order to contrast successful versus unsuccessful rollouts. The following table (discounted returns) shows that CPL underperforms in goal-reaching quality (the results were included in Table 3 of the revised manuscript).
>
> **Table:** Discounted average returns on Maze2D tasks. Numbers are averaged over 5 seeds; each evaluation uses 50 trajectories (1 = successful goal contact).
> | Environment | UMaze | Medium | Large |
> |-------------|-------|--------|--------|
> | **ReBRAC**  | 2.07  | **0.71** | 0.34  |
> | **CPL**     | 1.19  | 0.54     | 0.34  |
> | **CDE**     | 1.05  | 0.57     | 0.55  |
> | **PREFORL** | **2.22** | 0.67  | **0.63** |
>
> For a more interpretable comparison, the following table reports raw success rates under the no–goal-reset setting, where each trajectory terminates immediately after reaching the sampled goal. Across all three Maze2D datasets, PREFORL consistently surpasses CPL by a substantial margin, demonstrating that PREFORL achieves more reliable goal-directed behavior.
>
> **Table:** Success rates of CPL and PREFORL on D4RL Maze2D datasets.
> | Dataset    | CPL (succ.) | PREFORL (succ.) |
> |------------|--------------|------------------|
> | Umaze   | 82%          | **96.6%**        |
> | Medium  | 48%          | **66.3%**        |
> | Large-v1   | 26%          | **55.0%**        |

---

### Official Review · Reviewer_oreG · 2025-11-03

**Soundness:** 2
**Presentation:** 3
**Contribution:** 2
**Rating:** 4
**Confidence:** 4

**Summary:**

This paper proposes PREFORL, a preference-based offline reinforcement learning method that addresses value overestimation in sparse-reward settings. The approach trains policies via contrastive learning between successful demonstrations and synthetic degraded trajectories generated through action perturbation or state-based substitution. Core contributions include: A degradation framework augmenting sparse offline datasets, (2) A preference optimization loss bypassing explicit value estimation, and (3) Theoretical analysis linking the loss to policy imitation. Evaluations on Adroit, Sparse-MuJoCo, Maze2D, and MetaWorld benchmarks show PREFORL outperforms offline RL/imitation baselines in success rates and normalized scores.

**Strengths:**

- Novel degradation framework towards both action and state level.
- Comprehensive experimental validation.
- Complete theoretical analysis.
- Good writing for easy understanding.

**Weaknesses:**

- More navigation tasks (e.g. Antmaze-umaze/medium/large-diverse/replay), as well as offline RL baselines, should be performed.
- The ablation study of the degraded dataset size is lacking.

**Questions:**

- The proposed paradigm includes both a plug-in data-augmentation pipeline and a corresponding contrastive training pipeline. Can this paradigm be implemented on other BC-based offline-RL methods like Decision Transformer[1]?  I am glad to see how this paradigm performs on the DT backbone with different scales of data and models.
- Why is the state-degradation dataset constructed with the nearest neighbor state instead of directly adding noise like the action-degradation? I think there should be a comparison study of these two different state-degradation manners.

[1] Decision Transformer: Reinforcement Learning via Sequence Modeling

---

> ### Author Response · Authors · 2025-11-26
>
> We appreciate your insightful feedback and constructive comments!
>
> **The ablation study of the degraded dataset size is lacking.**
>
> We performed an ablation study varying the size of the degraded dataset by changing the degraded-to-preferred ratio ($|\mathcal{D}^{\downarrow s} \cup \mathcal{D}^{\downarrow a}| / |\mathcal{D}^+|$) from 1 to 10, and also considered the limiting case where negatives are generated on-the-fly ($\infty$) during training loops. As shown in the following table (included in the revised manuscript in Table 5 in Appendix D), performance remains stable across all settings: the normalized scores vary by less than 1–2 points on every Adroit task, and no consistent trend emerges as the ratio increases. This indicates that PREFORL is insensitive to the size of degraded samples and does not require a large or exhaustively constructed degraded dataset.
>
> **Table:** Impact of the degraded-to-preferred sample ratio on Adroit tasks. Scores are normalized and averaged over 3 seeds; ∞ denotes on-the-fly degraded sample generation.
> | Task            | Ratio = ∞ | Ratio = 1 | Ratio = 5 | Ratio = 10 |
> |-----------------|-----------|-----------|-----------|------------|
> | door-expert     | 106.0     | 106.1     | 105.2     | 105.9      |
> | hammer-expert   | 128.6     | 128.3     | 128.4     | 128.3      |
> | pen-expert      | 144.8     | 143.1     | 144.0     | 143.3      |
> | relocate-expert | 111.1     | 111.0     | 109.3     | 111.1      |
>
> This empirical robustness is consistent with our theory. The theoretical condition in Lemma 3.1 requires that the noise distribution has full support over a local neighborhood in the action space—not that the dataset enumerates this support exhaustively. Gaussian perturbation already satisfies this condition. In practice, drawing a modest number of perturbed trajectories provides sufficient "probabilistic coverage" to estimate the expectation in the preference loss.
>
> **Can this paradigm be implemented on other BC-based offline-RL methods like Decision Transformer?**
>
> PREFORL does not currently extend to Decision Transformers (DTs) with the same theoretical guarantees. Our theoretical framework assumes a direct correspondence between the optimal advantage function and the optimal policy, $A^\ast(s,a) = \alpha \log \pi^\ast(a\mid s)$, which allows PREFORL to estimate the optimal policy purely from local preference pairs. This relationship is well-defined for Markovian policies conditioned on state. In contrast, DTs condition on trajectory contexts and return-to-go, and thus does not factor into a Markovian $\pi(a\mid s)$. Consequently, the theoretical foundation underlying Lemma 3.1 does not directly extend to DTs.
>
> During the author response period, we nevertheless conducted experiments to explore whether PREFORL can be meaningfully combined with a DT-style architecture. We instantiated $\pi_\theta$ as a standard DT and constructed preference pairs by matching each expert trajectory segment $\varsigma^+\in\mathcal{D}^+$ with a degraded counterpart $\varsigma^-$ obtained by adding small Gaussian noise to the actions while retaining the same state and RTG (return-to-go) sequences. For each pair, we computed a trajectory-level score as the discounted sum of per-step log-probabilities assigned by the DT. The PREFORL loss $\mathcal{L}\_{\text{PREFORL}}$ encourages the DT to assign higher likelihood to $\varsigma^+$ than to $\varsigma^-$, and we trained the model using the combined objective
> $\mathcal{L}\_{\text{BC}} + \lambda \mathcal{L}\_{\text{PREFORL}}$. However, we did not observe consistent improvements over the standard DT baseline. We believe this is expected: because DTs condition on long-horizon trajectory returns rather than local Markovian state-action structure, local perturbations do not produce the kind of locally informative preference signals required by Lemma 3.1. In other words, PREFORL's strength lies in exploiting local advantage differences, whereas DT's sequence-model architecture is optimized for global return prediction. Exploring PREFORL extensions to DTs remains an interesting direction for future work.

---

> > ### Author Response · Authors · 2025-11-26
> >
> > **Why is the state-degradation dataset constructed with the nearest neighbor state instead of directly adding noise like the action-degradation?**
> >
> > We conducted an ablation study to empirically validate this alternative choice of state-based degradation. The last column of the following table reports the performance of PREFORL-StateNoise, which performs state degradation by directly adding Gaussian noise to states. As shown, this variant consistently performs worse across nearly all Adroit tasks—often dramatically so (e.g., hammer-cloned drops from 28.4 to 3.9).
> >
> > **Table:** Normalized scores of PREFORL compared to ReBRAC, CDE, CPL, and PREFORL-StateNoise (state-noise degradation variant) on D4RL Adroit tasks.
> > | Task            | ReBRAC          | CDE    | CPL           | PREFORL              | PREFORL-StateNoise   |
> > |-----------------|-----------------|--------|----------------|-----------------------|-----------------------|
> > | **pen-human**        | 103.5±14.1     | 72.1   | 100.1±2.2      | **119.1±3.1**         | 102.9±0.6             |
> > | **pen-cloned**       | 91.8±21.7      | 42.1   | 91.2±2.2       | 92.0±3.3              | **92.8±4.1**          |
> > | **pen-expert**       | **154.1±5.4**  | 105.0  | 130.9±3.2      | 144.8±3.1             | 139.8±4.1             |
> > | **door-human**       | 0.0±0.0        | 7.7    | 11.9±0.8       | **15.5±3.2**          | 14.5±5.5              |
> > | **door-cloned**      | 1.1±2.6        | 0.1    | 3.6±3.5        | **16.3±0.7**          | 2.9±1.2               |
> > | **door-expert**      | 104.6±2.4      | 105.9  | 105.8±0.2      | **106.0±0.0**         | 105.6±0.2             |
> > | **hammer-human**     | 0.2±0.2        | 1.9    | 15.1±8.7       | **16.6±3.0**          | 13.8±3.7              |
> > | **hammer-cloned**    | 6.7±3.7        | 7.3    | 13.2±8.1       | **28.4±3.2**          | 3.9±2.8               |
> > | **hammer-expert**    | **133.8±0.7**  | 126.3  | 128.3±0.3      | 128.6±0.2             | 127.7±0.3             |
> > | **relocate-human**   | 0.0±0.0        | 0.3    | 0.6±0.0        | **0.9±0.3**           | 0.9±0.4               |
> > | **relocate-cloned**  | 0.9±1.6        | 0.2    | 0.5±0.1        | **0.9±0.1**           | 0.2±0.1               |
> > | **relocate-expert**  | 106.6±3.2      | 102.6  | 110.2±0.4      | **111.2±0.7**         | 111.2±0.2             |
> >
> > These empirical findings align closely with our theoretical analysis. Our state-based degradation strategy is necessary because Lemma 3.1's identifiability result requires perturbing only the actions while keeping the state sequence fixed. Under Gaussian action perturbations, driving the preference loss to zero forces the learned advantage $\hat{A}(s,a)$ over $\mathcal{D}_{\text{pref}}$ to match the true advantage $A^\ast(s,a)$. Consequently, the learned policy satisfies $ \pi(\cdot \mid s) $ converges to $\pi^{\ast}(\cdot \mid s) $ for $ s \sim d^{\ast} $ where $d^\ast$ is the optimal state marginal (Line 93). This reasoning hinges on the fact that paired trajectory segments differ *only* in actions, so each segment-level preference constraint decomposes into local constraints $ \hat{A}(s,a) - \hat{A}(s,a') = A^{\ast}(s,a) - A^{\ast}(s,a') $. If noise were injected into the states, the resulting constraints would relate $ A^{\ast}(s,a) $ to $ A^{\ast}(s',a') $ for different states $ s \neq s' $. In this setting, many different advantage functions produce identical preferences over noisy trajectories while inducing different policies on the original state distribution $ d^{\ast} $. Then, the preference loss would no longer be sufficient to identify $ \pi^{\ast} $, and the conclusion of Lemma 3.1 would not hold. Thus, our state-based degradation is not arbitrary; it is necessary to obtain a rigorous convergence guarantee.

---

> > > ### Author Response · Authors · 2025-11-26
> > >
> > > **More navigation tasks as well as offline RL baselines should be performed.**
> > >
> > > Thank you for the suggestion to evaluate additional navigation tasks. Due to resource constraints, we selected a subset of the AntMaze suite where existing offline RL baselines, particularly IQL and ReBRAC, are less effective according to their results.
> > >
> > > **Table:** Average scores on selected **AntMaze** tasks. PREFORL results averaged over 3 seeds.
> > > | Dataset            | BC            | TD3+BC         | AWAC          | CQL           | IQL           | ReBRAC          | DT            | PREFORL            |
> > > |--------------------|---------------|----------------|---------------|---------------|---------------|------------------|---------------|---------------------|
> > > | **umaze-diverse-v2**  | 47.3±4.1      | 44.8±11.6      | 54.8±8.0      | 37.3±3.7      | 54.3±5.5      | **83.5±7.0**     | 51.8±0.4      | 68.9±3.2           |
> > > | **large-play-v2**     | 0.0±0.0       | 0.0±0.0        | 0.0±0.0       | 20.8±7.3      | 42.0±4.5      | 52.3±29.0        | 0.0±0.0       | **53.2±14.2**       |
> > > | **large-diverse-v2**  | 0.0±0.0       | 0.0±0.0        | 0.0±0.0       | 20.5±13.2     | 30.3±3.6      | **64.0±5.4**     | 0.0±0.0       | 26.1±7.1           |
> > >
> > > PREFORL achieves competitive performance on two of the three tasks. Large-play-v2 is the setting where PREFORL provides the clearest benefit: demonstrations in “play” datasets are noisy and locally inconsistent. PREFORL relies on local preference signals rather than global return estimation, allowing it to extract reliable supervision even when the offline data are highly suboptimal. PREFORL is less effective on large-diverse-v2. This dataset contains highly heterogeneous demonstrations that traverse many disconnected regions of the maze with less state overlap. As a result, the local preference pairs used by PREFORL become sparse in the parts of the state space relevant for reaching the goal, making it difficult for the model to propagate local improvements across the entire maze. In contrast, ReBRAC performs well on this task because it leverages global value-function regularization that benefits from the broad coverage in the diverse dataset.
> > >
> > > We appreciate the reviewer's suggestion to include more baselines. Our goal was to benchmark PREFORL against the strongest offline RL methods. Our main evaluations include ReBRAC, CDE, TD3+BC, IQL, and CQL, which collectively represent the top-performing families of modern offline RL. ReBRAC is the strongest reported method. To further strengthen the comparison, we incorporated three more baselines AWAC, SAC-N, and Decision Transformer (DT). This expanded set covers all major paradigms in offline RL, from value-based to policy-regularized to sequence-model methods, and includes every algorithm known to achieve state-of-the-art results on the tasks we evaluate. The results confirm that PREFORL remains competitive with or outperforms existing methods.
> > >
> > > **Table:** Adroit Tasks Performance (ReBRAC, AWAC, SAC-N, DT, PREFORL)
> > > | Task-Name         | ReBRAC             | AWAC                 | SAC-N              | DT                   | PREFORL                |
> > > |-------------------|--------------------|-----------------------|---------------------|-----------------------|-------------------------|
> > > | **pen-human**      | 103.5 ± 14.1       | 76.65 ± 11.71         | 6.86 ± 5.93         | 67.68 ± 5.48          | **119.1 ± 3.1**         |
> > > | **pen-cloned**     | 91.8 ± 21.7        | 85.72 ± 16.92         | 31.35 ± 2.14        | 64.43 ± 1.43          | **92.0 ± 3.3**          |
> > > | **pen-expert**     | 154.1 ± 5.4        | **159.91 ± 1.87**     | 87.11 ± 48.95       | 116.38 ± 1.27         | 144.8 ± 3.1             |
> > > | **door-human**     | 0.0 ± 0.0          | 2.39 ± 2.26           | -0.38 ± 0.00        | 4.44 ± 0.87           | **15.5 ± 3.2**          |
> > > | **door-cloned**    | 1.1 ± 2.6          | -0.01 ± 0.01          | -0.33 ± 0.00        | 7.64 ± 3.26           | **16.3 ± 0.7**          |
> > > | **door-expert**    | 104.6 ± 2.4        | 104.57 ± 0.31         | -0.33 ± 0.00        | 104.87 ± 0.39         | **106.0 ± 0.0**         |
> > > | **hammer-human**   | 0.2 ± 0.2          | 1.01 ± 0.51           | 0.24 ± 0.00         | 1.28 ± 0.15           | **16.6 ± 3.0**          |
> > > | **hammer-cloned**  | 6.7 ± 3.7          | 1.27 ± 2.11           | 0.14 ± 0.09         | 1.82 ± 0.55           | **28.4 ± 3.2**          |
> > > | **hammer-expert**  | **133.8 ± 0.7**    | 127.08 ± 0.13         | 25.13 ± 43.25       | 117.45 ± 6.65         | 128.6 ± 0.2             |
> > > | **relocate-human** | 0.0 ± 0.0          | 0.45 ± 0.53           | -0.31 ± 0.01        | 0.05 ± 0.01           | **0.9 ± 0.3**           |
> > > | **relocate-cloned**| 0.9 ± 1.6          | -0.01 ± 0.03          | -0.01 ± 0.10        | 0.16 ± 0.09           | **0.9 ± 0.1**           |
> > > | **relocate-expert**| 106.6 ± 3.2        | 109.52 ± 0.47         | -0.36 ± 0.00        | 104.28 ± 0.42         | **111.2 ± 0.7**         |

---

### Author Response · Authors · 2025-12-03
**Author Response Summary for Area Chair**

We thank all reviewers for their thoughtful feedback. Overall, the reviews positively acknowledge the novelty of our degradation framework for contrastive preference optimization in offline RL, the clarity of the writing, and the breadth of experiments. Below we concisely summarize how the revision addresses the concerns raised by each reviewer.

**1. Experimental Completeness (Navigation, Sensitivity Studies, Baselines)**

Reviewers requested additional benchmarks and ablations. New results added in the revision include:

* AntMaze Navigation tasks (Reviewer oreG): PREFORL performs competitively on large-play-v2 and large-diverse-v2, and is particularly effective on large-play. Compared to large-diverse, the large-play dataset has narrower and less coverage of the maze, which challenges value-based offline RL because the value function must extrapolate to regions with limited data support. PREFORL avoids global value prediction and instead uses local preference signals where data coverage is present, mitigating extrapolation issues in less-covered parts of the maze.
* Maze2D comparison (Reviewer aGGo): We introduced unsuccessful trajectories to enable a direct comparison with CPL. PREFORL substantially outperforms CPL in both returns and success rates.
* MetaWorld baselines (Reviewer uvFa): We incorporated CPL and ReBRAC on MetaWorld tasks using 50 demonstrations per task. PREFORL consistently achieves 90–100% success, while CPL and ReBRAC frequently fail.

These additions demonstrate that PREFORL is robust across multiple datasets, and forms of supervision.

* Hyperparameter robustness: We substantially expanded ablations to address all hyperparameter-related concerns:
  * Degraded-to-preferred dataset ratio (oreG): Varying the ratio from 1 to 10 (plus on-the-fly generation, ∞) shows <1–2 point variation on Adroit tasks.
  * Contrastive bias λ (k6nG and uvFa): Performance is stable across λ = 0.25, 0.5, and 1.0.
  * Segment length k (uvFa): Performance is similarly stable for k ∈ {20, 50, 100, 150}.
  * Noise variance σ (aGGo): Empirical results (as well as our theoretical analysis) show that as long as σ remains in the small-noise regime, success rates are stable and do not require environment-specific tuning.

Collectively, these ablations show that PREFORL is stable and insensitive to hyperparameters, directly addressing concerns about robustness.

In the rebuttal, we also provided broader offline RL baseline coverage (Reviewer oreG): We included AWAC, SAC-N, and Decision Transformer. PREFORL remains competitive or superior to these additional baselines.

**2. State-Based Degradation**

Reviewer oreG questioned the design choice of nearest neighbor state degradation versus Gaussian noise.
* Empirically, a new state-noise variant (Gaussian perturbation to states suggested by oreG) performs consistently worse — sometimes severely (e.g., hammer-cloned drops from 28.4 → 3.9).
* Theoretically, we clarified that perturbing states breaks the per-state advantage comparison required by Lemma 3.1. Identifiability requires trajectories differ only in actions while keeping state sequences fixed.

We also addressed Reviewer aGGo’s concern regarding computation overhead of nearest-neighbor state degradation.  We clarified that this degradation step is performed once offline with approximation. For larger datasets, we discuss amortized alternatives (conditional behavior models) that scale without brute-force search.

**3. Theory Clarification and Proof Rigor**

Lemma 3.1 states that, under local Gaussian perturbations of actions, driving the preference loss to zero forces the learned policy to match the optimal policy on the optimal state distribution. To address concerns about the clarity and rigor of Lemma 3.1 (Reviewer k6nG), we revised the proof of Lemma 3.1 to remove informal approximations and provide a cleaner argument for identifiability under local action perturbations. The proofs explains why preference learning avoids overestimation in existing offline RL methods: instead of estimating global Q-values, driving the preference loss to zero forces the learned advantage differences to match the true local advantage differences, which prevents assigning high probability to actions that lie outside the dataset's support.

We revised the introduction to provide citations supporting sensitivity of existing offline RL methods to support mismatches and sparse rewards (Reviewer uvFa).

**4. Decision Transformer (DT) Discussion**

Reviewer oreG requested experiments combining PREFORL with DT. During the author response period, we implemented a DT-style architecture with a PREFORL loss, but observed no consistent improvement. Theoretically, DT conditions on full-trajectory returns, whereas PREFORL’s analysis assumes a Markovian local structure with localized action perturbations. This clarifies the scope of our guarantees and outlines promising future directions.

We thank the AC for reviewing our submission!

---

### Meta-Review · Area_Chair_pMco · 2026-01-02

**Summary:**

To address the prevalent problem of value overestimation in sparse-reward offline reinforcement learning, this paper proposes the PREFORL framework. By conducting contrastive preference learning between successful demonstrations and synthetic degraded trajectories, this method enables policy learning without the need for explicit value estimation. Reviewers consistently agreed that the introduced mechanism, which employs two controlled degradation operators to generate negative samples, possesses significant novelty. Furthermore, the method has demonstrated performance superior to existing baselines across multiple challenging benchmarks, including Adroit, Sparse-MuJoCo, and MetaWorld. Given that the authors have properly addressed the reviewers' concerns, making the paper more complete in terms of both theory and experiments, I am inclined to accept this paper

**Reviewer Concerns:**

**Resolved Reviewer Comments**

**Experimental Completeness:** Addressing concerns regarding insufficient experimental validation, the authors added the challenging AntMaze tasks and multiple SOTA baselines, compellingly demonstrating the method's robustness and superiority in sparse-reward and low-coverage settings.

**State-Based Degradation Design:** Addressing concerns regarding the design of the degradation mechanism, the authors demonstrated through comparative experiments that simple state noise leads to significant performance deterioration. Furthermore, they theoretically clarified the necessity of keeping states fixed to guarantee the identifiability of the advantage function, thereby validating the rationality of the existing design.

**Parameter Sensitivity Analysis:** Addressing concerns regarding hyperparameter robustness, the authors conducted extensive ablation studies on noise variance, degraded data ratios, and key hyperparameters. The results indicate that the algorithm is insensitive to parameter variations and does not require fine-tuning for specific environments.

**Theory Clarification and Proof Rigor:** Addressing concerns regarding the theoretical proofs, the authors refined the details of the proof for Lemma 3.1 in the revised version.

**Unresolved Reviewer Comments**

**Implementation on Decision Transformer (DT):** Following reviewer suggestions regarding extensibility, the authors attempted to apply PREFORL to the Decision Transformer (DT) but observed no consistent performance improvements. While this reveals a limitation regarding non-Markovian sequence models, it does not undermine the method's core contributions and effectiveness in standard offline RL tasks.

**Reviewer Scores:**

Given that some reviewers did not participate in the subsequent discussion or provide further feedback after the authors' response, based solely on the initial scores and the quality of the rebuttal, the estimated score for this paper ranges from 6 to 8. In their response, the authors provided compelling supplementary experiments and theoretical clarifications, significantly enhancing the rigor of the paper and robustly demonstrating the effectiveness of the method.

---

### Decision · Program_Chairs · 2026-01-26

Accept (Poster)